# Cache What Lasts: Token Retention for Memory-Bounded KV Cache in LLMs

**Ngoc Bui[Y,J]\***, **Shubham Sharma[J]**, **Simran Lamba[J]**, **Saumitra Mishra[J]**, **Rex Ying[Y]**

[Y]Department of Computer Science, Yale University, [J]JPMorganChase AI Research

{ngoc.bui,rex.ying}@yale.edu
{shubham.x2.sharma,simran.lamba,saumitra.mishra}@jpmchase.com

**Source Code:** https://github.com/ngocbh/trimkv

## ABSTRACT

Memory and computation remain core bottlenecks in long-horizon LLM inference due to the quadratic cost of self-attention and the ever-growing key-value (KV) cache. Existing strategies for memory-bounded inference, such as quantization, offloading, or heuristic KV eviction, either incur high orchestration costs or rely on unreliable attention-based proxies of importance. We propose TRIM-KV, a novel approach that learns each token's intrinsic importance at creation time via a lightweight retention gate. Each gate predicts a scalar retention score that decays over time, reflecting the long-term utility of the token for a specific layer and head. Tokens with low scores are evicted when the memory budget is exceeded, ensuring that the cache always contains the most critical tokens. TRIM-KV is trained efficiently through distillation from a frozen LLM combined with a capacity loss, requiring only gate fine-tuning and adding negligible inference overhead. Across mathematical reasoning (GSM8K, MATH-500, AIME24), procedural generation (LongProc), conversational long-memory benchmarks (LongMemEval), and long-context understanding (LongBenchV2 and SCBench), TRIM-KV consistently outperforms strong eviction and learnable retrieval baselines, especially in low-memory regimes. Remarkably, it even surpasses full-cache models in some settings, showing that selective retention can serve as a form of regularization, suppressing noise from uninformative tokens. Qualitative analyses further reveal that learned retention scores align with human intuition, naturally recovering heuristics such as sink tokens, sliding windows, and gist compression without explicit design. Beyond efficiency, retention scores provide insights into layer- and head-specific roles, suggesting a new path toward LLM interpretability.

## 1 INTRODUCTION

Modern large language models (LLMs) can, in principle, handle extremely long input contexts – some recent models support context windows of 128k tokens or more (Yang et al., 2025; Gao et al., 2024). Yet, extending context length comes with steep computational costs. The self-attention mechanism has quadratic time complexity in sequence length, and storing the key-value (KV) cache for thousands of tokens can quickly exhaust GPU memory (Wang et al., 2025; Li et al., 2024a). In practical deployments, the KV cache, which saves past key and value vectors to avoid re-computation, becomes a major memory and latency bottleneck for long-context inference. Decoupling resource usage from context length is therefore critical for enabling efficient and scalable applications such as long-horizon reasoning (Chen et al., 2025) and lifelong agents (Zheng et al., 2025; Li et al., 2024d).

To address this challenge, recent work has explored memory-bounded LLMs that can operate effectively under constrained KV budgets (Li et al., 2024a). One line of research focuses on compression and quantization, aiming to reduce memory footprint by learning compact representations of past tokens rather than storing all keys and values explicitly (Hooper et al., 2024; Saxena et al., 2024).

---

*Work done during an internship at JPMorganChase AI Research

These techniques are mostly effective during the prefill phase but scale poorly with generation length. Another line leverages attention sparsity to offload most of the cache to CPU or secondary storage, and retrieve only relevant segments on demand via similarity search (Tang et al., 2024) or learned indices (Gao et al., 2025). While offloading lowers the on-GPU footprint, it incurs nontrivial orchestration overhead that accumulates over long generations, undermining end-to-end throughput.

A more common and direct approach to enforce a fixed memory budget is KV cache eviction, which directly drops certain tokens from the KV cache (Xiao et al., 2023). Many KV eviction strategies have been proposed to decide which tokens to remove. However, most of them are attention-guided heuristics: they track attention from new queries to cached tokens and retain those that are recently or frequently attended, adapting the cache to the current focus (Zhang et al., 2023; Li et al., 2024c; Wang et al., 2025; Liu et al., 2025; Ghadia et al., 2025; Cai et al., 2025). While being efficient, these methods assume that recent attention is a reliable proxy for future importance. This assumption often breaks for long-horizon generation and reasoning tasks: a token might be crucial much later, even if it has not been attended to in the recent past (Jiang et al., 2024). Moreover, attention-based eviction can suffer from attention bias, *e.g.,* the model might temporarily overlook a needed token due to a distracting context (Shi et al., 2023), causing it to be evicted prematurely. While some recent studies have attempted to learn better eviction decisions (Chen et al., 2024; Zeng et al., 2024), these methods typically scale poorly with sequence length and are therefore limited to the prefilling stage.

**In this work**, we take a new perspective on the KV eviction problem. Rather than relying on the attention-guided importance, we propose to learn each token's intrinsic importance at the time of its creation and use that as the basis for eviction. Intuitively, not all tokens are created equal: some carry significant semantic or task-related weight (*e.g.* a critical fact, a question being answered, or the first few "sink" tokens that often encode the topic or instructions), while others are relatively inconsequential (*e.g.* filler words, stopwords, or trivial arithmetic steps in a chain-of-thought). Moreover, the importance of tokens is not uniform across the network, but it varies systematically by layers and heads, reflecting their functional specializations (Voita et al., 2019; Wu et al., 2024b).

We posit that the contextual embedding of a token already encodes much of its long-term utility. We therefore introduce a retention gate that maps the token's embedding and produces a scalar retention score $\beta \in [0, 1]$ reflecting the token's inherent importance for a specific layer and head. Especially, we design this retention score to decay exponentially as the context grows, mimicking the gradual forgetting of old information in human brains (Ebbinghaus, 2013). Thus, a highly important token will have $\beta \approx 1$ and retain a high score for a long time, whereas a token deemed unimportant will have $\beta$ closer to 0 and its influence will vanish quickly. We leverage this score to drive a simple eviction policy: whenever the number of cached tokens exceeds the budget $M$, we evict the token with the smallest current retention score. This approach, which we call **T**oken **R**etent**I**on for **M**emory-bounded **KV** Cache (TRIM-KV), ensures that at all times, the cache is filled with the $M$ tokens judged most intrinsically important, with a preference toward more recently generated tokens.

Implementing retention-based caching in an existing LLM only requires adding a few lightweight components. We integrate the retention gates into each self-attention layer of a pretrained model to modulate attention weights by token importance during training. We then train only the gates with a two-part loss: a distillation loss that compels the modified model to mimic the original model's outputs, thus preserving quality, and a capacity loss that penalizes exceeding the target memory budget, thus encouraging sparseness in attention via eviction. Importantly, by training the gates across all layers jointly, the model can learn a coordinated, globally optimal caching policy rather than greedy layer-wise decisions. At inference time, the learned retention gates produce per-token scores on the fly, and eviction is implemented with a simple score comparison, adding minimal overhead.

**Results and Contributions.** Through extensive experiments on long-context and long-generation benchmarks, we demonstrate that our learnable token retention approach substantially improves the performance of memory-bounded LLMs. On challenging mathematical reasoning datasets, GSM8K, MATH, AIME, a long procedural generation benchmark, LongProc, and a long-memory chat assistant benchmark, LongMemEval, our method consistently outperforms eviction baselines, even when those baselines use $4\times$ more KV budget, and deliver 58.9% relative gain on pass@1 compared to the SOTA learnable KV retrieval baseline (Gao et al., 2025), especially in low-memory regimes. Remarkably, in several settings, TRIM-KV even surpasses a full-cache model, suggesting that selective retention can function as an effective regularizer by suppressing noise from uninformative tokens.

We also present qualitative evidence that learned retention scores align with human intuition: the model tends to assign high scores to initial tokens and problem descriptions, and low scores to less meaningful punctuation. Notably, many behaviors reminiscent of common heuristics, such as keeping sink tokens, sliding windows, and gist compression (Mu et al., 2023), emerge naturally and adaptively from our learned policy, without being hard-coded. Finally, we show that these learned retention scores can also act as a diagnostic tool for probing layer- and head-specific dynamics, providing a lightweight means to analyze and ultimately improve the interpretability of attention patterns.

## 2 RELATED WORK

**KV Cache Compression.** As model sizes and context windows grow, optimizing KV-cache memory is increasingly critical. Prior work largely falls into three directions: (i) token eviction/merging (Xiao et al., 2023; Li et al., 2024c; Zhang et al., 2023; Nawrot et al., 2024; Zhang et al., 2024; Qin et al., 2025; Wang et al., 2025; Liu et al., 2025; Park et al., 2025; Cai et al., 2025; Park et al., 2025; Kim et al., 2024), (ii) vector compression/quantization (Hooper et al., 2024; Liu et al., 2024b; Yue et al., 2024; Sun et al., 2024a), and (iii) token retrieval (Tang et al., 2024; Liu et al., 2024a; Gao et al., 2025). While effective in many settings, vector compression and retrieval either discard fine-grained information or introduce nontrivial systems overhead (e.g., coordination and data movement) (Li et al., 2024a). Moreover, their memory and computation still scale with sequence length, making them inefficient for long-horizon generation applications. Token eviction offers a simple, memory-bounded alternative; however, most existing policies are heuristic and can significantly degrade performance, especially on long reasoning trajectories. Recent work has introduced learnable eviction policies (Chen et al., 2024; Zeng et al., 2024; Huang et al., 2024), but these are primarily designed for the pre-filling stage and thus are not well suited to sustained long-horizon generation. We bridge this gap by introducing a learnable and efficient eviction policy designed for long-horizon LLM inference under fixed memory budgets.

**Forgetting in Language Models.** A key limitation of vanilla self-attention is the lack of an explicit forgetting mechanism, forcing the model to carry potentially irrelevant information and making long-context processing inefficient. Early work tackled this by replacing quadratic attention with linearized and recurrent variants (Katharopoulos et al., 2020; Wang et al., 2020; Sun et al., 2023; Yang et al., 2023; 2024) that summarize the past into a fixed-size state, often a single vector. While computationally attractive, such heavy compression can degrade performance on tasks requiring long-range memory. Follow-up studies (Behrouz et al., 2024; Sun et al., 2024b; Karami et al.; Karami and Mirrokni, 2025) increase memory capacity by replacing this hidden vector with a more expressive neural state. Complementary lines of work retain softmax attention but enforce forgetting by modifying attention logits (Lin et al., 2025) or imposing trainable sparsity patterns (Yuan et al., 2025). However, these approaches typically alter attention dynamics substantially and thus require training models from scratch. This incurs significant training cost and leaves their scalability to contemporary LLM sizes uncertain. In contrast, we introduce a *plug-in* forgetting mechanism for pretrained LLMs that converts them into memory-bounded models, providing long-context efficiency without retraining from scratch.

## 3 PRELIMINARIES

### 3.1 TRANSFORMERS WITH SELF-ATTENTION

Given a sequence of $d$-dimensional input vectors $\mathbf{x}_1, \ldots, \mathbf{x}_T$, a (causal) self-attention layer attends only to past positions. For each $t = 1, \ldots, T$, the attention output $\mathbf{o}_t$ is computed as

$$\mathbf{q}_t = \mathbf{W}_Q \mathbf{x}_t, \mathbf{k}_t = \mathbf{W}_K \mathbf{x}_t, \mathbf{v}_t = \mathbf{W}_V \mathbf{x}_t, \quad \mathbf{o}_t = \sum_{i=1}^{t} \frac{\exp\left(\mathbf{q}_t^\top \mathbf{k}_i\right)}{\sum_{j=1}^{t} \exp\left(\mathbf{q}_t^\top \mathbf{k}_j\right)} \mathbf{v}_i,$$

where $\mathbf{q}, \mathbf{k}, \mathbf{v}$ are query, key, and value states, respectively, and $\mathbf{W}_Q, \mathbf{W}_K, \mathbf{W}_V \in \mathbb{R}^{d \times d}$ are linear transformation weights. Here, we assume a single-head attention layer and omit the scaling factor $1/\sqrt{d}$ for simplicity. The sequence of key-value pairs $\{(\mathbf{k}_i, \mathbf{v}_i)\}_i$ is the in-context memory of the LLM. During the autoregressive decoding, we typically generate one token at a time and cache the running key-value pair $(\mathbf{k}_t, \mathbf{v}_t)$ to our in-context memory to avoid recomputation. However, this

vanilla caching approach leads to a linear increase in memory footprint with the sequence length, while computation grows quadratically (Keles et al., 2023). This reduces efficiency when handling long-context inputs and extended generation tasks.

## 3.2 REVISITING KV CACHE EVICTION

A common method to address the linear growth in the memory is to prune or compress the running key-value pairs into fixed-size (slot) memory. As new tokens arrive, we evict *un-(or less-)important* tokens from our memory and append the new ones. To understand this procedure, we revisit and rewrite the attention computation with eviction at inference step $t$ as follows:

$$\mathbf{o}'_t = \sum_{i=1}^{t} \frac{\alpha_{ti} \exp\left(\mathbf{q}_t^\top \mathbf{k}_i\right)}{\sum_{j=1}^{t} \alpha_{tj} \exp\left(\mathbf{q}_t^\top \mathbf{k}_j\right)} \mathbf{v}_i \quad \text{where} \quad \alpha_{ti} \in \{0,1\} \text{ and } \alpha_{ti} \geq \alpha_{t+1,i}, \ \forall i, t. \tag{1}$$

In Equation (1), we introduce a binary variable $\alpha_{ti} \in 0, 1$ indicating whether key–value pair $i$ has been evicted at time $t$ and the monotonicity constraint $\alpha_{ti} \geq \alpha_{t+1,i}$ ensures that we cannot retrieve a token once it is evicted (Figure 1). The goal is to choose a decision variable $\alpha$ so that the attention output deviates as little as possible from the full KV cache (all $\alpha_{ti} = 1, \ \forall i, t$).

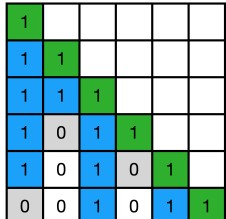

$$\min_{\alpha} \ \mathcal{L}_{\text{base}}(\mathbf{o}'_t; \mathbf{o}_t) \quad \text{s.t.} \sum_{i=1}^{t} \alpha_{ti} \leq M. \tag{2}$$

Figure 1: Attention w/ eviction ($M = 3$).

Here, $\mathcal{L}$ penalizes differences between attention with and without eviction, and the constraint enforces keeping at most $M$ tokens at any inference step $t$.

Solving the above constrained optimization at every time step $t$ is impractical due to its combinatorial nature and efficiency requirements of LLM inference in real-world applications. Most existing approaches (Xiao et al., 2023; Han et al., 2023; Zhang et al., 2023; Li et al., 2024c; Cai et al., 2025; Ghadia et al., 2025) opt to determine $\alpha$ heuristically while we focus on a learnable eviction method.

## 4 METHODOLOGY

In this section, we propose a learning-based eviction policy that prunes the KV cache based on the *intrinsic importance* of the tokens at each layer and head. The policy ranks tokens by relative importance to decide which should be evicted from the KV memory. To learn token importance, we introduce a small neural network that takes token embeddings as input and produces a scalar retention score. We then integrate this retention score into the attention computation to modulate the attention weights. We term this proxy attention mechanism a *retention-gated attention*. We train the LLM with retention-gated attention against a baseline model with standard attention, using a combination of distillation and hinge-like regularization losses to enforce memory capacity constraints while preserving response quality. A visualization is shown in Figure 2.

### 4.1 SELECTIVE IN-CONTEXT MEMORY VIA RETENTION-GATED ATTENTION

We introduce retention-gated attention, a trainable mechanism that mimics the information loss induced by inference-time eviction. From the formulation (1), the sequence $\alpha_{ii}, \alpha_{(i+1)i}, \ldots, \alpha_{ti}$ represents how token $i$ is retained in the attention computation over time. Retention begins at $1$ and then abruptly drops to $0$ once the token is evicted. While this binary behavior matches the inference stage, it poses challenges for learning: the signal is discrete, non-differentiable, thus providing no gradients for optimization. To remedy this, we replace the hard binary variable $\alpha$ with a smooth, monotonically decreasing function that models the gradual decay of importance while enabling gradient-based training. A natural candidate is the sigmoid function, $\bar{\alpha}_{ti} = 1/(1 + \exp(f(\mathbf{x}_i, t)))$, which predicts the time at which the token is evicted. However, this design suffers from two drawbacks: (i) the domain of $f$ is unnormalized since the sequence length is unknown during decoding, and (ii) the sigmoid flattens across most of its range, producing negligible variation between steps and leading to vanishing gradients during training.

To overcome these limitations, we adopt an exponential decay formulation, $\bar{\alpha}_{ti} = \beta_i^{t-i}$ where $\beta_i \in [0,1]$, to model the retention rate of token $i$ over time. Larger values of $\beta_i$ correspond to higher intrinsic importance, implying slower decay and stronger memory retention. Substituting this design

for $\alpha$ in Equation (1) yields our proposed *retention-gated attention*:

$$\mathbf{q}_t = \mathbf{W}_Q\mathbf{x}_t, \mathbf{k}_t = \mathbf{W}_K\mathbf{x}_t, \mathbf{v}_t = \mathbf{W}_V\mathbf{x}_t, \beta_t = g(\mathbf{x}_t), \quad \mathbf{o}_t = \sum_{i=1}^{t} \frac{\beta_i^{t-i}\exp\left(\mathbf{q}_t^\top\mathbf{k}_i\right)}{\sum_{j=1}^{t}\beta_j^{t-j}\exp\left(\mathbf{q}_t^\top\mathbf{k}_j\right)}\mathbf{v}_i. \quad (3)$$

Here, we propose a *retention gate* $g$, which is a lightweight network, to parametrize the token importance $\beta_t$. The retention gate can be a linear projection, *i.e.*, $g(\mathbf{x}) = \sigma(\mathbf{W}_\beta\mathbf{x}_t + b), \mathbf{W}_\beta \in \mathbb{R}^{1\times d}$, or a simple MLP, *i.e.*, $g(\mathbf{x}) = \sigma(\mathrm{MLP}(\mathbf{x}) + b)$. The sigmoid function $\sigma$ squashes the output of $g$ to the range $[0, 1]$, while $b$ is a learnable bias. When all $\beta_t = 1, \forall t$, our retention-gated attention recovers the vanilla attention. By incorporating the retention score into the exponential term, the attention weight can be written as $\exp(\mathbf{q}_t^\top\mathbf{k}_i + (t-i)\log\beta_i)$, which reveals that the retention score acts as an additive bias on the attention logits (Press et al., 2021).

**Token Retention vs. Attention Scores.** In standard self-attention, the importance of a past token $i$ at decoding step $t$ is given by $a_{ti} \propto \exp(\mathbf{q}_t^\top\mathbf{k}_i)$, which depends explicitly on the *current* query $\mathbf{q}_t$. These scores capture *short-term* utility for predicting the next token and are recomputed at every step, making them local, myopic, and highly dependent on the transient decoding state.

KV cache eviction, in contrast, is a *long-horizon* decision: once a token is dropped, it cannot influence *any* future prediction. An effective eviction policy should depend on a token's *intrinsic long-term utility* that reflects how useful it is expected to be over the remainder of the sequence and how long it has already stayed in the cache, rather than on the current query alone.

Token retention provides a more suitable abstraction. Instead of asking "how much should token $i$ contribute *now*?" it asks "how important is token $i$ for the long run, and for how long should it stay in the cache?" Concretely, each token $i$ receives a scalar retention score $\beta_i \in [0, 1]$ based only on information available at creation time (its representation, layer, and head), and its effective contribution at future step $t$ decays as $\beta_i^{t-i}$. This yields a smooth, exponentially decaying retention curve that is aligned naturally with long-term utility under memory constraints.

**Brain-inspired Interpretation.** Our retention-gated attention bears a natural connection to the Ebbinghaus's forgetting curve (Ebbinghaus, 2013), which models human memory retention as exponential decay, $R = \exp(-tS)$, where $S$ denotes memory strength (Woźniak et al., 1995).

Similarly, our retention-gated attention models each token's contribution as exponentially decaying with temporal distance, $\exp((t-i)\log\beta_i)$. Tokens start with full weight and gradually lose influence as new tokens arrive. The parameter $\beta$ acts analogously to memory strength: larger values yield more persistent, durable memories, while smaller values indicate weaker memories that fade quickly. This design embeds a forgetting mechanism into attention, enabling dynamic prioritization of recent or salient tokens while discarding less useful context, akin to human memory. Unlike prior work that applies the forgetting curve to retrieval-augmented generation (Zhong et al., 2024), we integrate it directly into the attention mechanism.

## 4.2 TRAINING

Our goal is to train the retention gate $g$ so that the LLM can preserve response quality under a memory constraint, thereby bridging the gap with the inference stage. Instead of training a separate gate for each layer and head, as formulated in Problem (2), we optimize all retention gates jointly in an end-to-end fashion. This holistic approach mitigates error propagation, allowing the model to learn a coordinated, globally optimal caching policy rather than greedy layerwise decisions. Starting from a pretrained LLM, we replace every standard attention block with our proposed retention-gated attention. Each

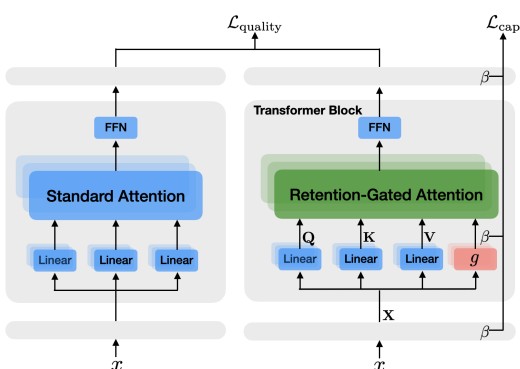

Figure 2: Training architecture.

block is equipped with a lightweight retention gate $g$ that maps token representations to retention scores $\beta_t \in [0, 1]$, which are then used to modulate attention weights according to Equation (3). We call this proxy LLM a retention-gated LLM.

**Objectives.** To train these retention gates, we formulate the training objective that balances two goals: (i) preserving the predictive quality of the original pretrained LLM, and (ii) enforcing memory capacity constraints by controlling the sum of retention scores at each step.

For the first objective, we use a combination of the distillation and standard next-token prediction losses. The distillation loss encourages the proxy LLM to align its output distribution with that of the baseline LLM using standard attention. In parallel, the next-token prediction loss enables the model to uncover sparsity patterns directly from the data, extending beyond the constraints of the pretrained LLM. Let $p(\cdot)$ and $q_\theta(\cdot)$ be the output distribution of the pretrained LLM and retention-gated LLM, respectively, where $\theta$ denotes the parameters of all retention gates. The quality loss is given by

$$\mathcal{L}_{\text{quality}} = \mathcal{L}_{\text{KL}} + \mathcal{L}_{\text{NTP}} = \mathcal{D}_{\text{KL}}\big(p(\cdot|x) \,\|\, q_\theta(\cdot|x)\big) + \mathbb{E}_{(x,y)}[-\log q_\theta(y|x)]. \quad (4)$$

Here, $\mathcal{D}_{\text{KL}}$ is the standard forward Kullback-Leibler divergence (Kullback and Leibler, 1951).

For the second objective, we impose a hinge-like regularization penalty, which discourages the model from exceeding the available KV memory slots at each step. For a retention gate within a given layer and KV head, the memory capacity loss is defined as:

$$\mathcal{L}_{\text{cap}} = \frac{1}{T} \sum_{t=1}^{T} \frac{1}{t} \max \{0, \sum_{i=1}^{t} \beta_i^{t-i} - M\}, \quad (5)$$

where $T$ is the sequence length and $M$ is the predefined memory capacity. Here, $M$ acts as a *soft* hyperparameter, primarily intended to prevent over-optimization during the early decoding stage when the sequence remains short. Training is performed with a fixed value of $M$, while inference can flexibly accommodate different KV budgets. This regularization is applied uniformly across all layers and KV heads of the transformer. The combined training objective is then:

$$\min_\theta \mathcal{L}_{\text{quality}} + \lambda_{\text{cap}} \mathcal{L}_{\text{cap}}, \quad (6)$$

where $\lambda_{\text{cap}}$ is a hyperparameter balancing between quality and capacity loss. Note that during training, only the retention gate parameters are updated, while all other model weights remain frozen.

**Hardware-aware Computation.** Retention-gated attention is fully parallelizable and compatible with FlashAttention-style kernels (Dao, 2023). We implement it with FlexAttention (Dong et al., 2024) plus a custom Triton kernel for the capacity loss $\mathcal{L}_{\text{cap}}$, performing forward/backward without materializing the full attention or $\beta$ matrices. This enables long-context training (up to 128K tokens on four H100 GPUs) with minor overhead versus standard parameter-efficient fine-tuning.

## 4.3 INFERENCE

At inference time, the base LLM is augmented with the retention gates learned during training (Section 4.2). These gates provide token-level intrinsic importance scores $\beta_i$, which quantify how strongly each past token should be retained for future computations. Unlike training, where the retention gates are used to modulate the attention weights, at inference, they act purely as decision-makers for eviction, operating alongside but independently of attention computation.

The eviction process is designed to ensure that the KV cache respects a predefined memory budget. Let $S_t \subseteq \{1, \ldots, t\}$ denote the set of tokens currently stored in the KV cache at decoding step $t$. When a new token $t + 1$ is generated, it is provisionally added to the cache. If this addition causes the cache size to exceed the memory capacity $M$, an eviction is triggered. The eviction rule is simple yet principled: we remove the token with the lowest retention score, i.e.,

$$j_{\text{evic}} = \arg\min_{j \in S_t} \{\beta_j^{t-j} | j \in S_t\}.$$

Intuitively, this favors retaining tokens deemed globally important by the learned retention gates while discarding those with little long-term value. In practice, this makes inference both memory-efficient and adaptive: as new context arrives, the model continually re-evaluates the importance of older tokens, enabling long-context generation while keeping memory usage bounded. Algorithm 1 illustrates a single decoding step, where attention computation is coupled with token eviction.

**Complexity.** Our inference is simpler and more efficient than existing works, including pure heuristic baselines (Li et al., 2024c). Table 6 shows that, at 32K context, TRIM-KV achieves $\sim 2\times$ higher decoding throughput than full-cache decoding and even faster than SnapKV, a purely heuristic method. More details are provided in Appendix A.2.

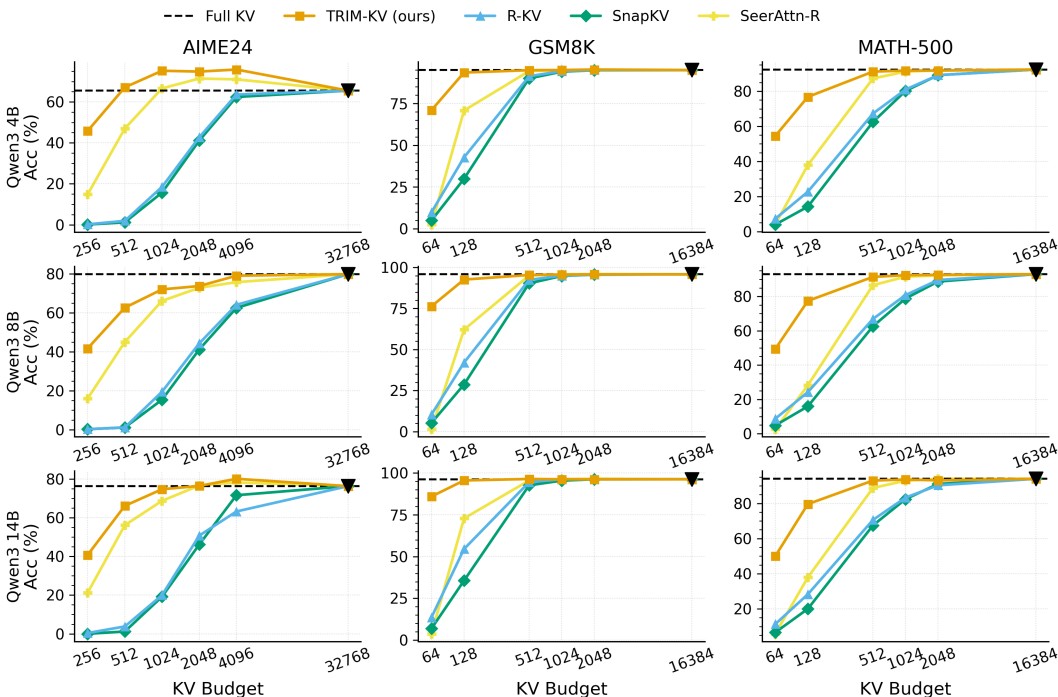

Figure 3: Patero frontiers of competing algorithms with different budgets on math benchmarks.

## 5 EXPERIMENTS

In this section, we conduct extensive experiments to demonstrate the performance advantages of our method on both *long-context* and *long-generation* tasks.

### 5.1 LONG GENERATION EVALUATION

**Benchmarks.** Following prior work (Gao et al., 2025; Cai et al., 2025), we evaluate on standard math-reasoning suites—AIME24 (Art of Problem Solving, 2024), GSM8K (Cobbe et al., 2021), and MATH-500 (Hendrycks et al., 2021). To assess performance beyond math reasoning and under long-context settings, we also report results on LongProc (Ye et al., 2025). Following (Gao et al., 2025), we report average pass@1 accuracy over 64 samples for AIME24 and 8 samples for GSM8K, MATH-500. We use greedy decoding for LongProc as the default in the benchmark.

**Base Models.** Following (Gao et al., 2025), we mainly use Qwen3's family models (Yang et al., 2025), including Qwen3-1.7B, Qwen3-4B, Qwen3-8B, Qwen3-14B and DeepSeek R1 Distill (Guo et al., 2025) variants including, DeepSeek-R1-Distill-Qwen-7B and DeepSeek-R1-Distill-Lllam-8B. We report the results with Qwen3 models in the main paper, and the remaining is in Appendix B.

**Baselines.** We compare our method against SeerAttn-R (Gao et al., 2025), R-KV (Cai et al., 2025), SnapKV (Li et al., 2024c), H2O (Zhang et al., 2023), StreamingLLM (Xiao et al., 2023). R-KV, SnapKV, H2O, and StreamingLLM are heuristic, recency-driven KV *eviction* policies for long-form generation under a fixed memory budget. SeerAttn-R is a learnable KV *retrieval* approach for reasoning tasks: rather than evicting, it offloads the full KV cache to host memory and uses recent queries to fetch relevant blocks for attention. KV retrieval methods preserve all past information but require nontrivial CPU–GPU orchestration and incur offloading overhead. We therefore treat SeerAttn-R as a strong learnable baseline, and R-KV/SnapKV as representative eviction baselines.

**Implementation Details.** We train the retention gates using OpenR1-MATH-220k (Hugging Face) dataset, similar to (Gao et al., 2025). Note that we only train the retention gates' weights while keeping the original model parameters frozen. We set the objective hyperparameters $\lambda_{\text{cap}} = 1.0$ and the memory capacity $M = 256$. Each transformer block has a retention gate $g$, which is a single

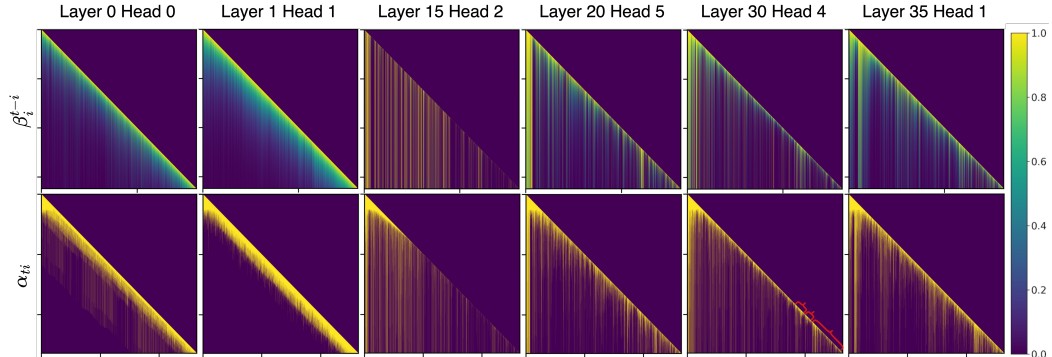

Figure 4: Visualization of token retention score $\beta_i^{t-i}$ (top) and eviction decisions $\alpha_{ti}$ (bottom).

MLP layer with the hidden dimension of $512$, thus having dimensions of $d \to 512 \to h$, where $h$ is the number of KV heads. We use the activation function as the default activation function in MLP layers of the base model. We initialize the bias in the retention gates to a large value (e.g., $b = 18.0$) to begin training with minimal forgetting or compression. All trainings are on 4 H100 80G GPUs.

### 5.1.1 QUANTITATIVE RESULT

**Math Reasoning Tasks.** Figure 3 shows our method outperforming all baselines by a large margin, especially in low-budget regimes. Notably, TRIM-KV surpasses attention-guided methods (R-KV, SnapKV) even when they are given $4\times$ KV budget. Under the same budget, *i.e.*$512$ for AIME24 and $128$ for GSM8K/MATH-500, it yields a **198.4%** relative improvement over these baselines. Against the SOTA learnable KV retrieval baseline, TRIM-KV outperforms SeerAttn-R across all settings, yielding a **58.9%** pass@1 gain at the same budget. Crucially, TRIM-KV operates in a pure KV-eviction regime, a stricter setting than the KV retrieval methods such as SeerAttn-R, and thus avoids CPU–GPU offloading overhead. In some settings, like for Qwen3-4B model and AIME24 dataset, our method can even surpass the standard full KV cache. These results suggest that a large fraction of KV-cache tokens in reasoning models is redundant and can be discarded without degrading performance.

**Long Procedural Generation Tasks.** We evaluate KV-eviction methods on tasks that require both long-context comprehension and extended generation. Table 1 reports results with Qwen3-4B model. Overall, TRIM-KV consistently outperforms all other eviction baselines and, in several settings, even surpasses the full-cache model. Moreover, this result highlights that TRIM-KV with retention gates trained on math-reasoning data generalizes well to non-math tasks. Full results and analysis are provided in Appendix B.

| Method$_{\text{KV budget}}$ | CountDown | | | Pseudo to Code | |
|---|---|---|---|---|---|
| | 0.5k | 2k | 8k | 0.5k | 2k |
| FullKV | 96.0 | 90.0 | **69.0** | 50.8 | **25.0** |
| StreamingLLM$_{2048}$ | 7.0 | 5.0 | 5.0 | 20.6 | 1.5 |
| H2O$_{2048}$ | 12.0 | 7.5 | 2.5 | 33.7 | 0.5 |
| SnapKV$_{2048}$ | 57.0 | 49.0 | 13.0 | 42.7 | 4.5 |
| R-KV$_{2048}$ | 88.5 | 81.0 | 63.0 | 48.2 | 2.5 |
| TRIM-KV$_{2048}$ | **96.5** | **93.5** | 66.5 | 49.3 | 18.0 |

Table 1: Qwen3-4B on LongProc. Bold is for the best, underline is for the best KV eviction.

### 5.1.2 QUALITATIVE RESULT

To examine the eviction policy learned by our retention gates, we run TRIM-KV on Qwen3-4B for the first example in AIME24. Figure 5**a**–**b** show the mean retention score, averaged over all layers and heads, for each token in the example sequence. Aligning with our intuition, retention gates assign high scores to task-relevant tokens (e.g., `ometer`, `shop`, `walk`, `minutes`) and to the initial token `<|im_start|>`, which often serves as an attention sink. In contrast, whitespace and punctuation receive low retention scores and are discarded early, yielding short lifespans in the KV cache. Next, we examine retention scores and eviction decisions at layer–head granularity.

**Emergent Eviction Heuristics.** Figure 4 visualizes the retention scores $\beta_i^{t-i}$ and eviction decisions $\alpha_{ti}$ for selected layers and heads. Many eviction heuristics, such as attention sinks (Xiao et al., 2023), sliding windows (Zhu et al., 2021), A-shape (Jiang et al., 2024), emerge naturally from our learned

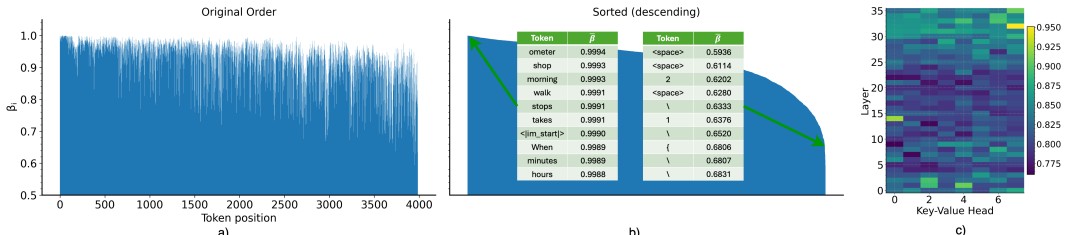

Figure 5: *a)* Average retention scores across all layers and heads of Qwen3-4B on tokens of an AIME24 example. *b)* Top 10 tokens with the highest (left table) and lowest (right table) average retention. *c)* The layer- and head-wise sparsity level estimated by token retentions.

| Method | En.MultiChoice | Retr.KV | ICL.ManyShot | Math.Find | En.QA | Code.RepoQA | En.Sum | Mix.Sum+NIAH | Retr.MultiHop |
|---|---|---|---|---|---|---|---|---|---|
| Full KV | 20.52 | 66.00 | 95.57 | 32.60 | 28.78 | 53.86 | 36.48 | 38.33 | 49.60 |
| StreamingLLM$_{4096}$ | 5.68 | **2.20** | **100.00** | 13.20 | _6.84_ | 2.96 | _29.21_ | 28.25 | 0.00 |
| H2O$_{4096}$ | 4.80 | 0.00 | **100.00** | 8.00 | 3.70 | 0.46 | 8.97 | 6.51 | 0.27 |
| SnapKV$_{4096}$ | _10.04_ | 0.00 | **100.00** | **18.60** | 6.29 | 0.23 | 27.90 | _29.28_ | _0.31_ |
| TRIM-KV$_{4096}$ | **23.58** | 0.00 | **100.00** | _13.40_ | **13.68** | **3.86** | **34.02** | **35.44** | **49.16** |

Table 2: Performance on long-context tasks from the SCBench benchmark.

policy without being hard-coded, and they adapt to the functional roles of individual layers and heads. For instance, sliding-window behavior is more common in early layers, whereas attention sinks appear more frequently in later layers (see Figure 11 and 12 for a comprehensive view). Moreover, TRIM-KV adapts the window size by layer and head: in *Layer 1/Head 1*, tokens receive nearly uniform retention scores, so the KV cache behaves like a recency-biased window that keeps the most recent tokens; in *Layer 0/Head 0*, multiple sliding windows of varying widths emerge from the learned policy; in *Layer 15/Head 2*, no sliding window is observed because certain tokens receive substantially higher retention than others, suggesting a specialized functional role for this head. The A-shaped pattern typically appears in layers that emphasize instruction/problem-statement tokens (*e.g.*, *Layer 20/Head 5* and *Layer 30/Head 4*) or chain-of-thought/reasoning prompts (*e.g.*, *Layer 35/Head 1*). These heads also exhibit context switching, where small, dense lower-triangular blocks emerge and then fade quickly when the context changes or a sentence completes. To the best of our knowledge, the absence of a sliding window, the presence of multiple coexisting windows, and context switching are newly observed eviction patterns that arise naturally from our learned policy.

**Token Retention Enables Interpretability.** Beyond guiding eviction policy, token-level retention scores provide a diagnostic tool for analyzing the functional roles of individual KV heads in the base LLM. Visualizing retention scores alongside the tokens preserved in the KV cache after generation reveals distinct specializations: some heads emphasize a recency window (Figure 14 and 17), whereas others preferentially retain mathematical tokens-numbers and operators (Figures 19), as well as problem-description tokens (Figures 16). Even function or filler words, such as pronouns, prepositions, conjunctions, `wait` and `let`, tend to be kept by specific heads (Figures 13). In particular, some heads tend to retain the period token while discarding others (Figures 18). We hypothesize that, in these heads, periods act as implicit *gist* tokens (Mu et al., 2023), summarizing the information in the preceding sentences.

Our analyses indicate that KV heads in LLM develop different functional roles and therefore keep different types of tokens. These tokens are often dispersed across the context rather than forming contiguous chunks, as each already captures contextual information. This observation contrasts with existing approaches (Yuan et al., 2025; Gao et al., 2025) that advocate chunk- or block-based KV-cache. Instead, we show that keeping a small number of high-context tokens is more budget-effective.

**Budget Allocation.** Figure 5c reports head- and layer-wise sparsity estimated from the retention scores *i.e.*, $1 - \frac{2}{T(T+1)} \sum_{i<t} \beta_i^{t-i}$. We observe that later layers are typically sparser than earlier ones, consistent with prior findings in (Cai et al., 2024). Practically, the retention scores enable heterogeneous budgets across KV heads under a global constraint by evicting tokens with low global retention. However, existing KV-cache and FlashAttention implementations assume uniform sequence lengths across heads within a layer; enabling efficient per-head variable-length caches is left to future work.

## 5.2 LONG-CONTEXT EVALUATION

**Long-context Decoding.** We evaluate TRIM-KV on LongMemEval$_S$ (Wu et al., 2024a) and SCBench (Li et al., 2024b), two 128K-token benchmarks for long-term memory and long-context KV compression. We use Qwen3-4B-Instruct (Qwen3, 2025) as the base model and train the retention gates on SynthLong (Lazarevich et al., 2025), BookSum (Kryściński et al., 2021), and Buddhi (Singhal, 2024) with $M = 1024$. All other experimental settings follow Section 5.1. As shown in Table 3 and Table 2, TRIM-KV significantly outperforms baselines on LongMemEval, maintaining a good performance with only 25% of the budget, while others degrade sharply. On SCBench, it remains competitive across most tasks, though all KV eviction methods struggle on incompressible retrieval tasks (e.g., Retr.KV, Code.RepoQA), consistent with (Li et al., 2024b). Overall, TRIM-KV shows strong performance in both long-context and long-generation settings, unlike prior work that targets only one stage. More details of this experiment are in Appendix B.2.

| Method$_{\text{KV budget}}$ | Acc |
|---|---|
| Full KV$_{131072}$ | 49.4 |
| StreamingLLM$_{32768}$ | 27.6 |
| SnapKV$_{32768}$ | 27.8 |
| TRIM-KV$_{32768}$ | **44.8** |
| StreamingLLM$_{16384}$ | 19.0 |
| SnapKV$_{16384}$ | 18.2 |
| TRIM-KV$_{16384}$ | **38.6** |
| StreamingLLM$_{8192}$ | 13.0 |
| SnapKV$_{8192}$ | 15.8 |
| TRIM-KV$_{8192}$ | **34.0** |

Table 3: Results on LongMemEval$_S$.

**Chunk Prefilling.** We follow Huang et al. (2024) to evaluate TRIM-KV on the chunk-prefill settings. We train Phi3-mini-128K (Abdin et al., 2024) on LongAlpaca (Chen et al., 2023) with $M = 512$. All experimental settings follow (Huang et al., 2024) for fair comparison, detailed in Section B.3. Table 10 shows that TRIM-KV outperforms the LocRet, a trainable KV eviction baseline, and even full KV inference, by a significant margin.

| Method | Acc | $\Delta$ (%) |
|---|---|---|
| Full KV | 28.79 | 0.00 |
| LocRet | 28.03 | -2.64 |
| TRIM-KV | 34.09 | +18.41 |

Table 4: Results on LongBench-V2.

## 5.3 ABLATION STUDIES

We ablate the objective by training the Qwen3-4B retention gates with different loss combinations and report AIME24 pass@1 at a 4096-token budget in Table 5. Both forward KL and next-token prediction perform well on their own, and their combination further improves accuracy. The memory capacity loss is essential for compression, and removing it leads to a sharp drop. We provide comprehensive ablations with reversed KL, generalization with different training datasets, gate's architecture, and other hyperparameters such as $M$ in Appendix B.4.

| Method$_{\text{KV budget}}$ | pass@1 |
|---|---|
| Full KV$_{32768}$ | 65.5 |
| TRIM-KV$_{4096}$ | **75.8** |
| (TRIM-KV $-\mathcal{L}_{\text{KL}}$)$_{4096}$ | 72.1 |
| (TRIM-KV $-\mathcal{L}_{\text{NTP}}$)$_{4096}$ | 72.5 |
| (TRIM-KV $-\mathcal{L}_{\text{cap}}$)$_{4096}$ | 42.9 |

Table 5: Objective ablation.

## 6 CONCLUSION AND FUTURE WORK

We introduced TRIM-KV, a learnable, retention-gated approach to KV cache management that prioritizes tokens by intrinsic importance rather than recent attention. By training lightweight gates with distillation and a capacity loss, our method enforces strict memory budgets with a simple and efficient eviction policy. Extensive experiments across math reasoning, long-procedural generation, and conversational long-memory benchmarks demonstrate that our method outperforms strong eviction and retrieval baselines, and sometimes even surpasses full-cache models. Analyses show that the learned retention scores align with human intuitions and reveal layer- and head-specific dynamics, offering a simple probe for interpretability.

**Future Work.** Our results indicate that retention-gated attention is an effective learnable proxy for approximating standard attention with eviction during inference. In the current work, however, we keep the backbone parameters frozen during training and still rely on standard attention at inference time. A natural next step is to replace standard attention with retention-gated attention and train the retention mechanism jointly with the attention layers during pretraining or post-training. This could allow the retention scores to better cooperate with the learned query, key, and value states, shaping the model's behaviors from the outset rather than optimizing retention on top of a fixed attention stack. Such a design would enable training objectives that explicitly trade off task performance and memory usage, potentially yielding models that are inherently memory-bounded without requiring any post hoc compression policy.

Besides, building on these results, we plan to extend retention gating to multimodal inputs, tool-calling applications, and develop adaptive budgets that allocate memory across layers, heads, and tasks to further improve both performance and efficiency.

DISCLAIMER

This paper was prepared for informational purposes by the Artificial Intelligence Research group of JPMorgan Chase & Co. and its affiliates ("JP Morgan") and is not a product of the Research Department of JP Morgan. JP Morgan makes no representation and warranty whatsoever and disclaims all liability, for the completeness, accuracy or reliability of the information contained herein. This document is not intended as investment research or investment advice, or a recommendation, offer or solicitation for the purchase or sale of any security, financial instrument, financial product or service, or to be used in any way for evaluating the merits of participating in any transaction, and shall not constitute a solicitation under any jurisdiction or to any person, if such solicitation under such jurisdiction or to such person would be unlawful.

ETHICS STATEMENT

This work aims to improve the efficiency of large language models by reducing their memory and computational footprint. Our method can make long-context reasoning more accessible by lowering hardware costs, which may democratize access to advanced LLM capabilities. However, efficiency improvements may also accelerate the deployment of LLMs in high-stakes or resource-limited settings where risks around misinformation, bias, or misuse persist. We stress that our method does not mitigate these broader societal risks and should be paired with ongoing efforts in safety, fairness, and responsible deployment.

REPRODUCIBILITY STATEMENT

We ensure reproducibility by providing detailed descriptions of the model architecture, training objectives, and evaluation benchmarks in the main text and appendix. Hyperparameters, training schedules, and implementation details are included for all experiments. All datasets we use are publicly available, and we will release code, model checkpoints, and scripts for training and evaluation upon publication. Together, these materials allow independent researchers to fully reproduce and verify our results.

The authors used large language models to help refine and polish the writing of this manuscript.

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

## A  METHODOLOGY

### A.1  INFERENCE ALGORITHM.

Algorithm 1 illustrates the attention computation with KV eviction using retention gates for a single decoding step. We mark the parts that are different from the standard attention computation in blue.

---

**Algorithm 1:** Attention computation with KV eviction (single decoding step)

---

**Input** : current hidden $\mathbf{x}_t$; KV cache $\mathbf{K}_{t-1}, \mathbf{V}_{t-1}, \mathbf{B}_{t-1}$ indexed by $S_{t-1}$; retention gate $g$
**Param** : capacity $M$;
**Output** : attention output $\mathbf{o}_t$; updated $(\mathbf{K}_t, \mathbf{V}_t, \mathbf{B}_t)$; updated index set $S_t$

```
// 1) Project to Q/K/V for the current token
```
1 $\mathbf{q}_t \leftarrow \mathbf{W}_Q \mathbf{x}_t$; $\quad \mathbf{k}_t \leftarrow \mathbf{W}_K \mathbf{x}_t$; $\quad \mathbf{v}_t \leftarrow \mathbf{W}_V \mathbf{x}_t$; $\quad \beta_t = g(\mathbf{x}_t)$;
```
// 2) Append current token to the KV cache
```
2 $\mathbf{K}_t \leftarrow \mathbf{K}_{t-1} \cup \mathbf{k}_t$; $\quad \mathbf{V}_t \leftarrow \mathbf{V}_{t-1} \cup \mathbf{v}_t$; $\quad \mathbf{B}_t \leftarrow \mathbf{B}_{t-1} \cup \beta_t$; $\quad S_t \leftarrow S_{t-1} \cup \{t\}$;
```
// 3) Compute attention over currently cached keys/values
//    (restricted to S_t)
```
3 $\mathbf{o}_t \leftarrow \text{FLASHATTN}(\mathbf{q}_t, \mathbf{K}_t, \mathbf{V}_t)$;
```
// 4) If capacity exceeded, evict the least important token
```
4 **while** $|S_t| > M$ **do**
5 $\quad j_{\text{evic}} \leftarrow \arg\min \{\beta_j^{t-j} | j \in S_t\}$;
6 $\quad$ Remove $\mathbf{K}_t[j_{\text{evic}}], \mathbf{V}_t[j_{\text{evic}}], \mathbf{B}_t[j_{\text{evic}}]$;
7 $\quad S_t \leftarrow S_t \setminus \{j_{\text{evic}}\}$;
8 **end**

---

**Positional Encoding in KV Cache Eviction.** Our retention mechanism is designed to be positional-encoding agnostic and does not add any extra recency bias beyond what is already present in the base model. The exponential decay in the retention-gated attention is a smooth approximation of the decay process from 1 to 0 of the standard attention with eviction, not to encode the positional information. Therefore, regardless of whether the base model uses absolute positions, RoPE, or no positional encoding, that information is already folded into $\mathbf{x}$, $\mathbf{q}$, and $\mathbf{k}$ in the standard attention (see Eq 1).

Implementation-wise, when using RoPE, we follow prior work (R-KV, SnapKV) and cache post-rotated keys, so the eviction is orthogonal to the positional embeddings used.

## A.2 COMPLEXITY

**Memory efficiency.** Like other KV-eviction schemes, TRIM-KV uses a fixed-size cache with $\mathcal{O}(M)$ slots, independent of sequence length $T$. For each token (per head), it stores a single scalar retention score $\beta_i$, adding only $\approx 1/d_h$ overhead, where $d_h$ is the dimension of the key and vector states, relative to the KV states, which is negligible in practice. Unlike R-KV (Cai et al., 2025), TRIM-KV does not store queries.

| Method | Context Length | Gen Length | Batch Size | Throughput (tok/sec) | Decode Time (s) |
|---|---|---|---|---|---|
| FullKV | | | | 68.44 | 59.84 |
| SeerAttn-R | 32786 | 1024 | 4 | 68.93 | 59.41 |
| SnapKV | | | | 124.67 | 33.00 |
| TRIM-KV | | | | **130.48** | **31.39** |
| FullKV | | | | 114.39 | 35.8 |
| SeerAttn-R | 16378 | 1024 | 4 | 100.45 | 40.77 |
| SnapKV | | | | **153.21** | **26.73** |
| TRIM-KV | | | | 151.04 | 27.11 |
| FullKV | | | | 138.97 | 58.94 |
| SeerAttn-R | 16378 | 1024 | 8 | 139.34 | 58.78 |
| SnapKV | | | | 244.60 | 33.49 |
| TRIM-KV | | | | **279.90** | **29.26** |

Table 6: Throughput and decoding time comparisons of different KV cache methods on a single H200 GPU. The memory budget $M$ is 1024.

**Computational efficiency.** For each generated token, TRIM-KV computes a scalar retention score $\beta_i$ via a lightweight MLP that can be fused with QKV projections; scores are cached and not recomputed each step. During compression, it applies a temporal discount (elementwise power) and evicts the argmin; both costs only $\mathcal{O}(M)$. This is cheaper than heuristics like R-KV, which require

key–key similarity scoring over the cache. Table 6 reports throughput and latency: at 32K context, TRIM-KV achieves $\sim 2\times$ higher decoding throughput than full-cache decoding and even faster than SnapKV, a purely heuristic method. SeerAttn-R does not provide any computation advantage over full cache model.

# B    ADDITIONAL EXPERIMENTS

## B.1    LONG GENERATION EVALUATION

We provide more comprehensive experiment details in this section.

**Experiment settings.** For the training, we set the maximum training length to be 16384. We train the retention gates with a learning rate of $2 \times 10^{-4}$ and a weight decay of 0.01. For math reasoning tasks, we follow SeerAttn-R (Gao et al., 2025) that uses OpenR1-Math-220K (Hugging Face) dataset, which has 564M tokens for training. During training, we use a batch size of 1 for each GPU, and gradient accumulation is set to 4. Other hyperparameters are set to the default in Huggingface Trainer. All training is conducted on 4 H100 GPUs.

**Benchmarks.** AIME24 (Art of Problem Solving, 2024), GSM8K (Cobbe et al., 2021), and MATH-500 (Hendrycks et al., 2021) are standard math reasoning benchmarks. LongProc (Ye et al., 2025) is a long-context benchmark of six procedural-generation tasks that require integrating dispersed information and producing structured long-form outputs (up to $\sim 8K$ tokens)—from extracting tables from HTML to executing multi-step search to build feasible travel itineraries. The suite spans varied access patterns (sequential vs. targeted retrieval), deductive reasoning demands, and search execution, enabling stress tests of long-range coherence and procedure following. Each task includes deterministic solution procedures and structured outputs, allowing rule-based evaluation (row-level F1 for HTML→TSV, unit tests for pseudocode→code, exact-match traces for Path/ToM, and validators for Countdown/Travel). To probe generation length, we use three difficulty tiers targeting 0.5K, 2K, and 8K output tokens.

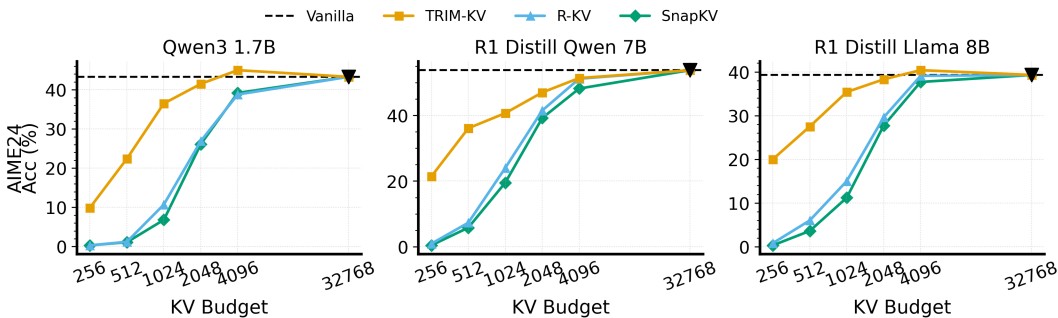

Figure 6: Patero frontiers of competing algorithms with different budgets on AIME24.

**Math reasoning results.** Figure 6 reports AIME24 performance for Qwen3-1.7B and DeepSeek-R1-Distill variants. Across both families, TRIM-KV consistently outperforms eviction baselines.

**Comparison to a query-agnostic baseline.** We provide a comparison to KeyDiff (Park et al., 2025), a query-agnostic baseline that only considers a key diversity for eviction. The result in Figure 7 shows that the performance of KeyDiff is significantly worse than that of other baselines. Note that R-KV can be considered as a generalization of KeyDiff since it considers both key diversity and attention scores for eviction heuristics.

**Results on LongProc.** Table 7 reports KV-eviction results on long procedure–generation tasks. Across tasks and budgets, TRIM-KV achieves the best performance, and it even surpasses the full-cache baseline on COUNTDOWN (0.5K/2K). Under tighter memory budgets, its margin over heuristic baselines widens.

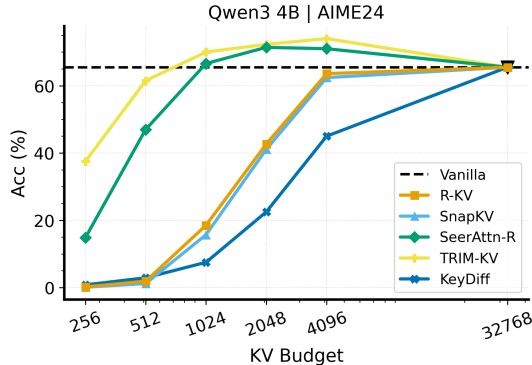

Figure 7: Comparison to KeyDiff, a query-agnostic baseline.

| Method$_{\text{KV budget}}$ | HTML to TSV | | | Thought of Mind | | | Travel Planning | |
| | 0.5k | 2k | 8k | 0.5k | 2k | 8k | 2k | 8k |
|---|---|---|---|---|---|---|---|---|
| FullKV | 49.0 | 41.6 | 13.9 | 33.0 | 10.5 | 0.0 | 0.0 | 0.0 |
| SnapKV$_{8192}$ | 37.1 | 9.3 | 0.1 | 26.0 | 7.0 | 0.0 | 0.0 | 0.0 |
| H2O$_{8192}$ | 28.3 | 6.4 | 0.4 | **38.0** | 7.0 | 0.0 | 0.0 | 0.0 |
| R-KV$_{8192}$ | 38.0 | 7.1 | 0.5 | 26.0 | 7.5 | 0.0 | 0.0 | 0.0 |
| TRIM-KV$_{8192}$ | **48.7** | **31.2** | **7.5** | 32.5 | **10.5** | 0.0 | 0.0 | 0.0 |
| StreamingLLM$_{2048}$ | 1.2 | 0.0 | 0.0 | 2.0 | 0.0 | 0.0 | 0.0 | 0.0 |
| SnapKV$_{2048}$ | 1.5 | 0.2 | 0.0 | 15.0 | 0.0 | 0.0 | 0.0 | 0.0 |
| H2O$_{2048}$ | 0.4 | 0.8 | 0.0 | 7.6 | 0.0 | 0.0 | 0.0 | 0.0 |
| R-KV$_{2048}$ | 1.6 | 0.1 | 0.0 | 3.0 | 0.0 | 0.0 | 0.0 | 0.0 |
| TRIM-KV$_{2048}$ | **14.7** | **3.9** | 0.0 | **18.0** | 0.5 | 0.0 | 0.0 | 0.0 |

Table 7: Results of Qwen3-4B across LongProc tasks: F1-score for HTML to TSV task and accuracies (%) for the remaining tasks. Best per task column in bold.

## B.2 Long-Context Evaluation

**Experimental settings.** We adopt Qwen3-4B-Instruct (Qwen3, 2025) as the base model, which supports a context window of up to 256K tokens. Retention gates are trained on a mixture of SynthLong-32K (Lazarevich et al., 2025), BookSum (Kryściński et al., 2021), and Buddhi (Singhal, 2024), covering sequence lengths from 32K to 128K tokens. We shuffle the combined corpus and train for 10,000 steps (i.e., 10,000 randomly sampled examples), with a maximum training sequence length of 128K and memory capacity $M = 1024$. All other settings follow Section 5.1.

**Benchmark Dataset.** We evaluate chat-assistant capabilities under strict memory budgets using LongMemEval$_S$ (Wu et al., 2024a). This subset provides contexts up to 123k tokens (measured with the Qwen3 tokenizer) and includes six question types that probe long-term memory: *single-session-user* (SS-User) and *single-session-assistant* (SS-Assist), which test recall of facts stated by the user or assistant within a session; *single-session-preference* (SS-Pref), which requires personalized responses from stored personal information; *multi-session* (Multi), which aggregates or compares information across sessions; *knowledge-update* (Knowledge), which tracks and answers with the most recent, changed user information; *temporal-reasoning* (Temporal), which reasons over timestamps and time references.

To evaluate KV-cache eviction methods on this benchmark, we follow the multi-turn, multi-session protocol of (Li et al., 2024b). Specifically, before each query, the eviction-based model must compress the accumulated dialogue into a fixed-size, reusable KV cache—mirroring real-world assistants that maintain state across turns and sessions under strict memory budgets. We use Qwen3-4B-Instruct (Qwen3, 2025) to assess whether model outputs match the ground-truth responses.

**Results.** The results in Table 8 show that our method outperforms baseline eviction strategies by a significant margin.

| Method$_{\text{KV budget}}$ | Overall | Multi | Knowledge | SS-User | SS-Pref | SS-Assist | Temporal |
|---|---|---|---|---|---|---|---|
| Full KV$_{131072}$ | 49.4 | 25.6 | 68.0 | 62.9 | 93.3 | 85.7 | 30.1 |
| StreamingLLM$_{32768}$ | 27.8 | 15.8 | 50.0 | 32.9 | 56.7 | 33.9 | 15.0 |
| SnapKV$_{32768}$ | 27.6 | 15.8 | 42.3 | 24.3 | 73.3 | 28.6 | 21.8 |
| TRIM-KV$_{32768}$ | **44.8** | **24.8** | **69.2** | **47.1** | **76.7** | **73.2** | **30.1** |
| StreamingLLM$_{16384}$ | 19.0 | 12.8 | 35.9 | 24.3 | 26.7 | 17.9 | 11.3 |
| SnapKV$_{16384}$ | 18.2 | 9.0 | 25.6 | 17.1 | 70.0 | 12.5 | 14.3 |
| TRIM-KV$_{16384}$ | **38.6** | **15.0** | **69.2** | **44.3** | **73.3** | **60.7** | **27.8** |
| StreamingLLM$_{8192}$ | 13.0 | 9.0 | 25.6 | 15.7 | 16.7 | 10.7 | 8.3 |
| SnapKV$_{8192}$ | 15.8 | 9.8 | 19.2 | 12.9 | 70.0 | 7.1 | 12.8 |
| TRIM-KV$_{8192}$ | **34.0** | **13.5** | **53.9** | **41.4** | **73.3** | **50.0** | **21.8** |
| StreamingLLM$_{4096}$ | 10.2 | 9.8 | 14.1 | 14.3 | 16.7 | 7.1 | 6.0 |
| SnapKV$_{4096}$ | 13.8 | 11.3 | 14.1 | 14.3 | 56.7 | 7.1 | 9.0 |
| TRIM-KV$_{4096}$ | **26.8** | **12.0** | **46.2** | **25.7** | **63.3** | **33.9** | **19.6** |

Table 8: Results on LongMemEval$_S$: overall and partial accuracies (%).

### B.3 CHUNKED-PREFILL EVALUATION

In this section, we evaluate our method in the chunked-prefill setting (Huang et al., 2024), which enables long-context inference without exceeding memory limits. In this framework, long prompts are split into multiple chunks; the model computes the KV cache for each chunk sequentially and compresses the cache whenever it surpasses the memory budget. We compare our method against LocRet (Huang et al., 2024), a learnable KV eviction baseline that also assigns token-importance scores for eviction. Following the experimental setup of LocRet, we evaluate TRIM-KV on the LongBench (Bai et al., 2024) and LongBench-V2 (Bai et al., 2025) benchmarks. For LongBench-V2, we restrict evaluation to examples with context length below 128K tokens, matching the maximum context length advertised for Phi3-mini-128K. To train TRIM-KV, we use only LongAlpaca (Chen et al., 2023), as in LocRet, to ensure that improvements are not attributable to differences in training data. In this setting, we set $M = 512$, and keep all other hyperparameters identical to those in Section 5.2.

For a fair comparison, we adopt the hyperparameters used by LocRet: the chunk size is set to 3072 and the budget size to 6000. We evaluate performance on Phi-3-mini-128K (Abdin et al., 2024), reproducing Table 6 from the LocRet paper. Since the original LongBench experiments have not been released, we use the default chat template of Phi-3-mini-128K for all runs. Table 9 and 10 report the results for LongBench (Bai et al., 2024) and LongBench-V2 (Bai et al., 2025) benchmarks, respectively. We observe discrepancies in the full-KV performance, likely due to differences in chat templates.

Overall, TRIM-KV remains highly effective in the chunked-prefill setting. On LongBench, TRIM-KV nearly matches full-KV performance, whereas LocRet exhibits a 4.82% drop relative to full-KV cache inference. Notably, on a more challenging benchmark, LongBench-V2, we even surpass the performance of the full KV cache by 6.5%.

Conceptually, LocRet is designed for chunk prefilling in long-context inference, while our method is general. The training paradigms also differ: LocRet predicts attention logits independently for each KV head, whereas we train retention gates end-to-end, allowing importance scores across heads to jointly optimize the eviction strategy. At inference time, LocRet further relies on a hand-crafted sliding window to preserve the most recent tokens from the latest chunk, and they show that removing this heuristic substantially degrades performance. In contrast, our method requires no such manually designed mechanism: sliding-window-like behavior emerges automatically from the learned policy when beneficial, and some heads (e.g., layer 15, head 2 in Figure 4) do not exhibit sliding-window patterns at all.

| Method | 2wikimqa | gov_report | hotpotqa | lcc | multi_news | mfieldqa | musique | narrativeqa | pssg_count | pssg_retrv | qasper | qmsum | repobench-p | samsum | triviaqa | Δ (%) |
|---|---|---|---|---|---|---|---|---|---|---|---|---|---|---|---|---|
| Full KV* | 33.37 | 33.35 | 43.06 | 51.86 | 26.57 | 49.82 | 19.82 | 18.21 | 2.97 | 93.50 | 41.07 | 19.51 | 58.02 | 23.15 | 86.38 | – |
| LocRet* | 35.93 | 33.46 | 48.70 | 52.61 | 26.41 | 52.77 | 25.12 | 24.56 | 3.00 | 83.00 | 40.17 | 23.35 | 57.16 | 26.37 | 82.39 | – |
| Full KV | 37.01 | 33.35 | 53.35 | 33.35 | 26.02 | 54.45 | 25.90 | 26.17 | 5.00 | 96.50 | 40.18 | 24.08 | 34.08 | 38.77 | 85.50 | 0.00 |
| LocRet | 37.24 | 32.80 | 48.67 | 28.60 | 26.77 | 54.12 | 26.63 | 22.96 | 3.50 | 87.50 | 39.39 | 22.98 | 37.28 | 37.99 | 83.29 | -4.82 |
| TRIM-KV | 36.86 | 33.45 | 52.84 | 32.97 | 26.07 | 53.92 | 26.12 | 23.77 | 4.00 | 89.00 | 40.17 | 23.96 | 34.74 | 38.74 | 85.25 | -2.54 |

Table 9: Performance on long-context tasks with Phi3-mini-128K on the LongBench benchmark, including average relative change (Avg. Δ) compared to Full KV. * indicates that the numbers are reported from (Huang et al., 2024, Table 6).

| Method | Acc. Short | Acc. Medium | Acc. Easy | Acc. Hard | Avg. Acc | Avg. Δ (%) |
|---|---|---|---|---|---|---|
| Full KV | 33.71 | 18.60 | 34.44 | 25.86 | 28.79 | 0.00 |
| LocRet | 32.02 | 19.78 | 26.67 | 28.74 | 28.03 | -2.64 |
| TRIM-KV | **35.39** | **31.39** | **37.78** | **32.18** | **34.09** | **+18.41** |

Table 10: Performance on long-context tasks of KV eviction methods with Phi3-mini-128K on the LongBench-V2 benchmark, including average relative change (Avg. Δ) compared to Full KV.

## B.4 ADDITIONAL ABLATION STUDIES

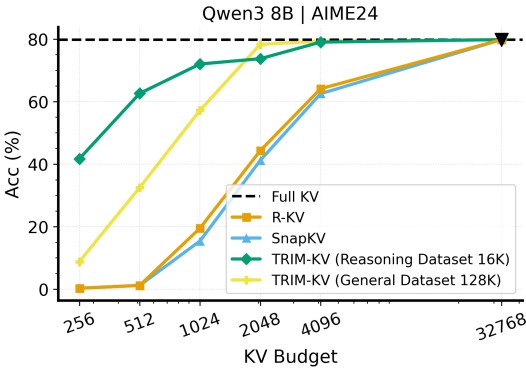

Figure 8: Generation Ablation.

**Ablation on training datasets.** In Section 5.1, we train the retention gates on a reasoning dataset—OpenR1-Math (Hugging Face)—and evaluate on AIME24, MATH-500, and GSM8K. This follows standard practice and matches the setting of (Gao et al., 2025), ensuring a fair comparison. To assess cross-domain generalization, we instead train the gates on general long-context datasets (SynthLong, BookSum, Buddhi), similar to Section 5.2, and then evaluate on math reasoning benchmarks to test whether retention scores learned from general data transfer to long chain-of-thought tasks. As shown in Figure 8, gates trained on general datasets generalize well and even surpass math-specific training at a 2048 KV budget. However, their performance degrades more quickly under tighter KV budgets. Overall, these results are promising and suggest that scaling the training of the retention gates by combining all datasets could yield further improvements.

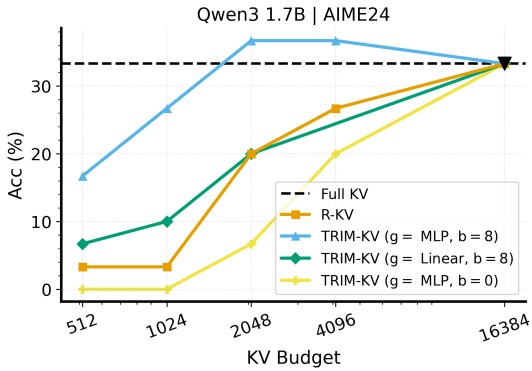

Figure 9: Ablating the retention gate's architecture.

**Ablation for the retention gate's architecture.** We evaluate several retention-gate architectures and report the performance of Qwen3 1.7B on AIME24 in Figure 9. Due to computational constraints, this ablation uses greedy decoding. For the MLP gate, we use a single-hidden-layer MLP with width 512. We find that the MLP gate outperforms a simple linear projection, and that a large positive initial bias is crucial for stable training by keeping the gate's output nearly 1 at initialization to ensure minimal early forgetting.

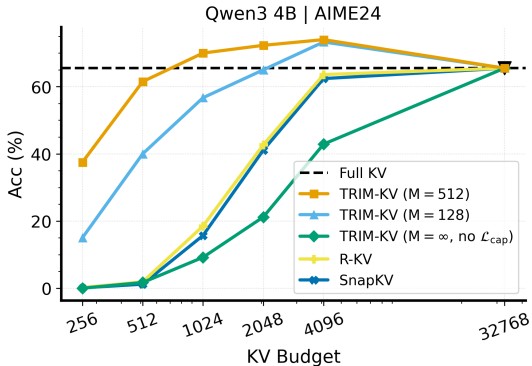

Figure 10: Ablating the training memory capacity $M$.

**Ablation on training memory capacity** $M$**.** We evaluate multiple settings of $M$ in Figure 10. With $M = \infty$, there is no capacity penalty, which hurts performance due to the absence of sparsity pressure. Setting $M = 128$ outperforms attention-guided heuristics but shows signs of over-optimizing for sparsity. In practice, we recommend choosing $M$ to match the expected deployment-time memory budget.

## C ADDITIONAL QUALITATIVE RESULTS

In this section, we provide more qualitative results to illustrate the eviction decisions made by TRIM-KV. All visualizations are from the first example in the AIME24 dataset. Please refer to Section 5.1.2 for discussions.

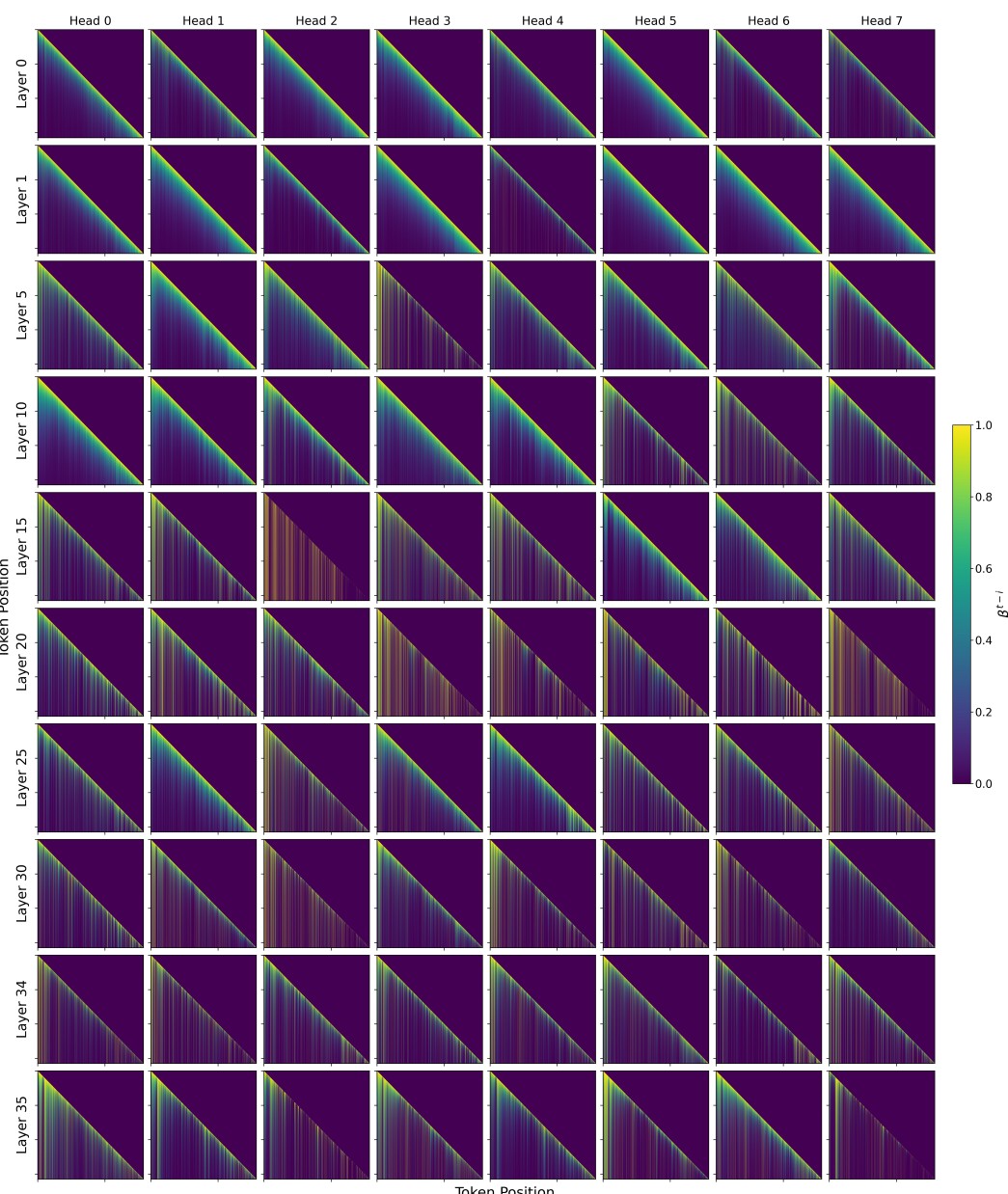

Figure 11: A visualization of token retention matrices of Qwen3-4B when answering a math question in the AIME24 dataset. Each subplot is a token retention matrix $\{\beta_i^{t-i}\}_i^t$ in a specific layer and head. **Observations:** earlier layers often exhibit sliding-window-like patterns, while later layers develop clearer functional specializations.

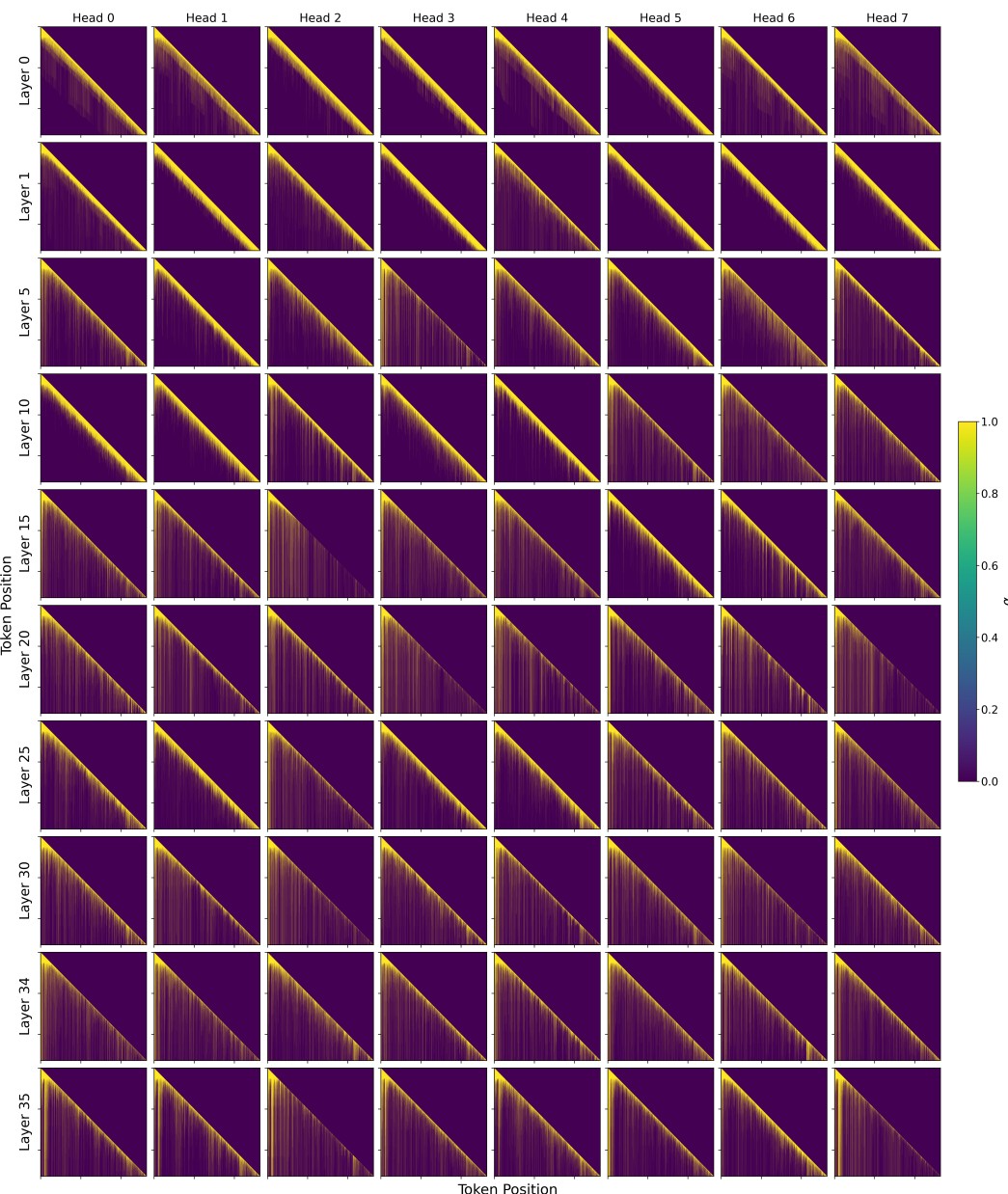

Figure 12: A visualization of eviction decisions of Qwen3-4B when answering a math question in the AIME24 dataset. Each subplot is a matrix of eviction decisions $\alpha_{ti}$ in a specific layer and head.

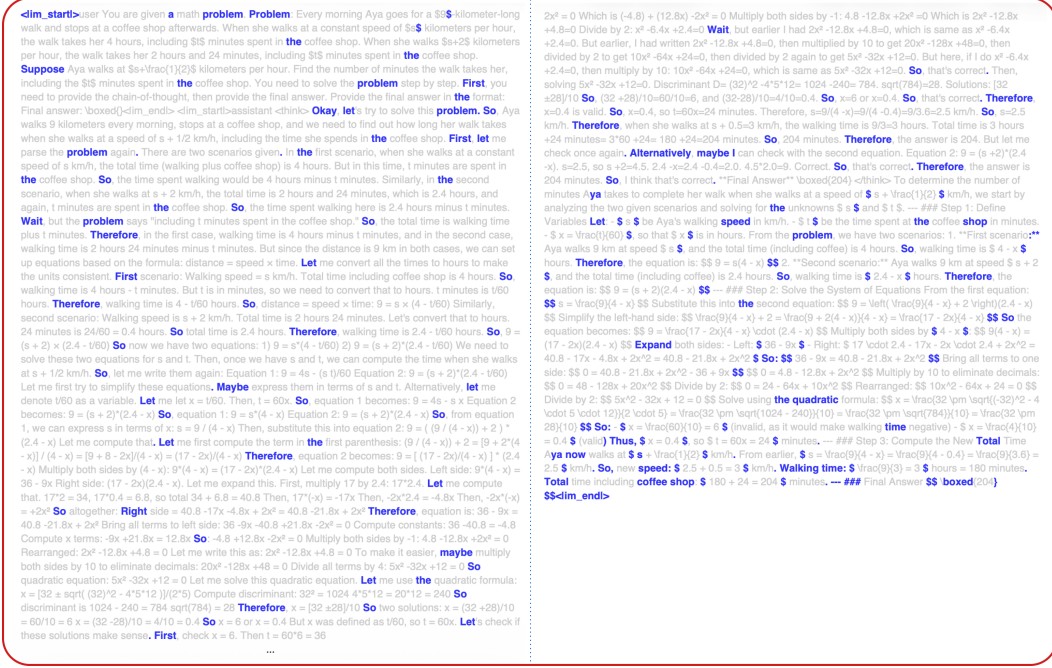

Figure 13: Visualization of tokens retained in the *layer 4 head 5* of the KV cache after generation, where the KV budget is 256. The model is Qwen3-4B. Bold blue indicates tokens retained in the KV cache, where gray indicates evicted tokens.

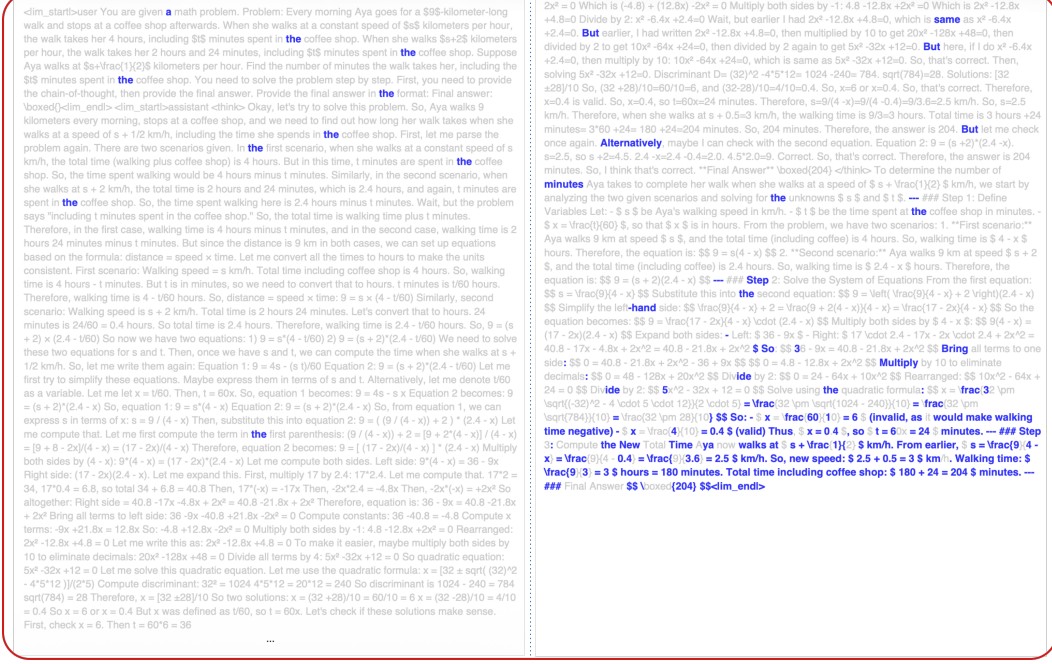

Figure 14: Visualization of tokens retained in the *layer 6 head 4* of the KV cache after generation, where the KV budget is 256. The model is Qwen3-4B. Bold blue indicates tokens retained in the KV cache, where gray indicates evicted tokens.

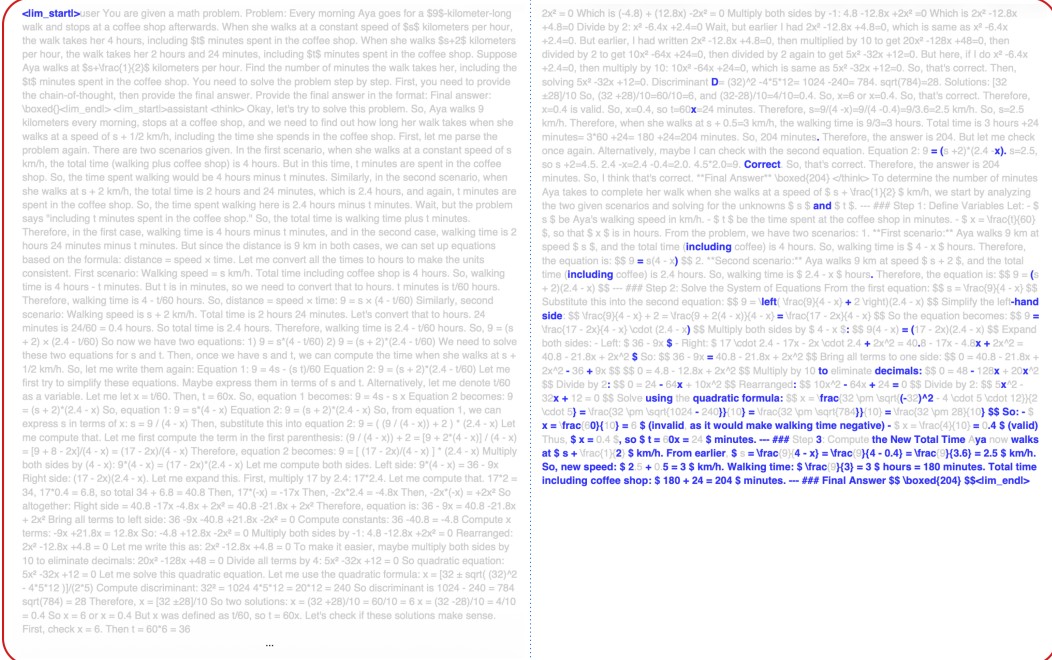

Figure 15: Visualization of tokens retained in the *layer 7 head 3* of the KV cache after generation, where the KV budget is 256. The model is Qwen3-4B. Bold blue indicates tokens retained in the KV cache, where gray indicates evicted tokens.

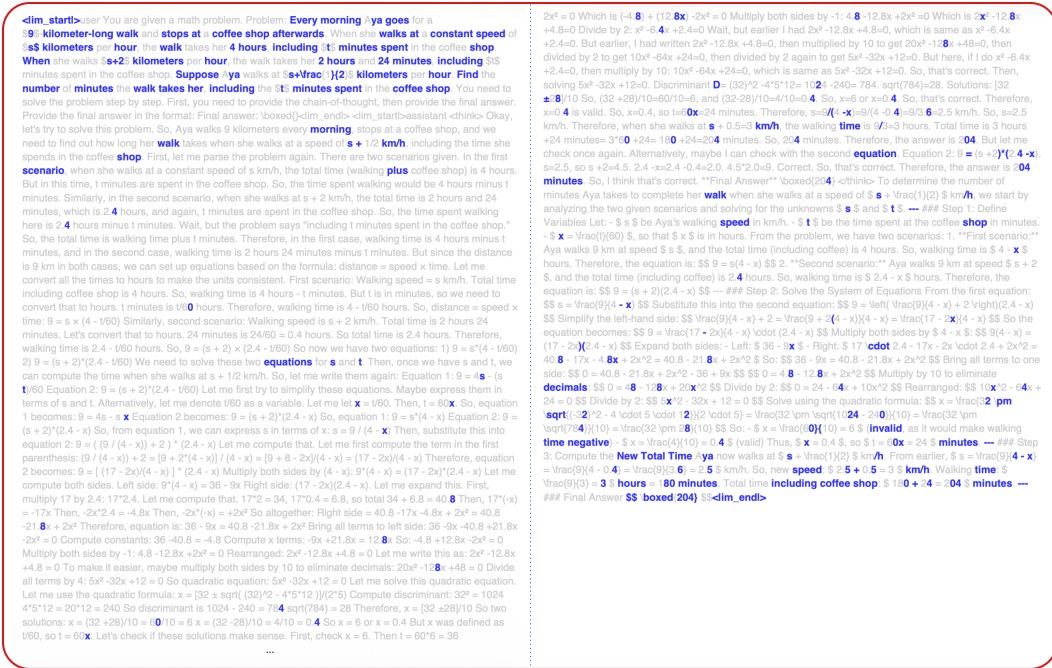

Figure 16: Visualization of tokens retained in the *layer 9 head 7* of the KV cache after generation, where the KV budget is 256. The model is Qwen3-4B. Bold blue indicates tokens retained in the KV cache, where gray indicates evicted tokens.

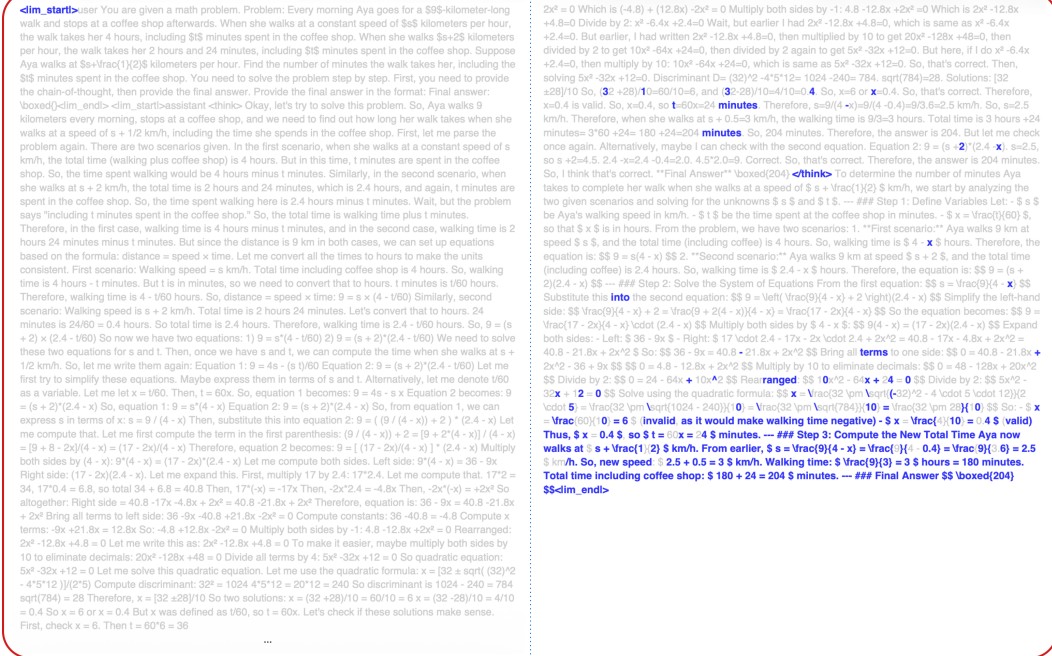

Figure 17: Visualization of tokens retained in the *layer 15 head 5* of the KV cache after generation, where the KV budget is 256. The model is Qwen3-4B. Bold blue indicates tokens retained in the KV cache, where gray indicates evicted tokens.

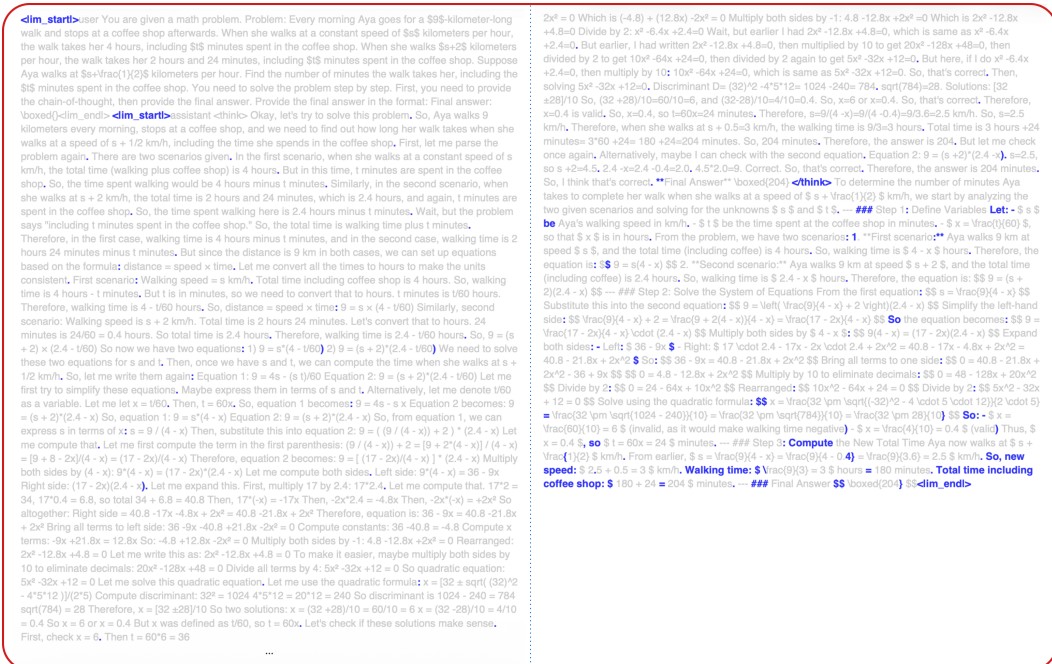

Figure 18: Visualization of tokens retained in the *layer 27 head 2* of the KV cache after generation, where the KV budget is 256. The model is Qwen3-4B. Bold blue indicates tokens retained in the KV cache, where gray indicates evicted tokens. **Observations:** This head mostly keeps period tokens. This suggests that this head may implicitly perform gist tokens that summarize information from the previous sentence. This contrasts with recent trends that advocate for saving a chunk of tokens (Yuan et al., 2025; Gao et al., 2025). Our results suggest that it can be more budget-efficient to save individual tokens because they already capture contextual information.

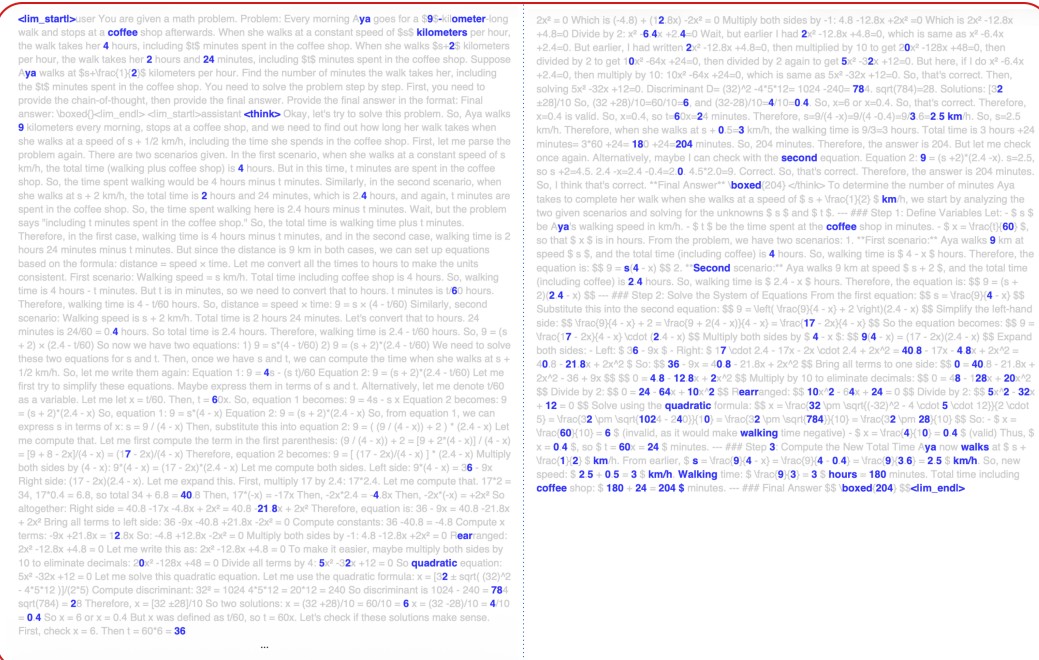

Figure 19: Visualization of tokens retained in the *layer 32 head 2* of the KV cache after generation, where the KV budget is 256. The model is Qwen3-4B. Bold blue indicates tokens retained in the KV cache, where gray indicates evicted tokens.

