# OpenReview forum: "Cache What Lasts: Token Retention for Memory-Bounded KV Cache in LLMs"
_ICLR.cc/2026/Conference — ICLR 2026 Poster_

### Official Review · Reviewer_aGD2 · 2025-10-16

**Soundness:** 4
**Presentation:** 4
**Contribution:** 4
**Rating:** 6
**Confidence:** 5

**Summary:**

This paper proposes TRIM-KV, a trainable sparse attention method for KV cache eviction. The core design of TRIM-KV is to learn a forgetting gate $\beta_i^{t-i}$ for the attention mechanism. Such gate is learned through minimizing the KL loss and language modeling loss while reducing the cap loss $L_{cap}$. A small MLP is applied to predict the gate value $\beta_i$ at each position, and only the MLP is trained while other parameters are frozen. KV cache eviction is conducted by evicting the KV pair with the minimal $\beta_i^{t-i}$. This paper trains such architecture on math reasoning datasets, and evaluate the trained model on various reasoning datasets that requires long cot generation. Empirical results show that TRIM-KV is able to outperform all the KV cache eviction baselines, and appears lossless acceleration in long-cot reasoning. The authors conduct detailed ablation studies to show the effectiveness of TRIM-KV's design and the generalizability across various training datasets.

**Strengths:**

1. The design of TRIM-KV is reasonable and effective. Learning a forgetting gate $\beta_i$ for KV cache eviction is a novel and interesting idea.

2. Experimental results show significant speedups, and TRIM-KV outperforms classical baselines such as SnapKV and R-KV. The ablation studies are detailed and comprehensive.

3. The paper is clearly written, making it easy for readers to understand both the training and inference procedures of TRIM-KV.

**Weaknesses:**

1. Since TRIM-KV evicts the KV cache according to query-agnostic features, i.e., the importance score of each KV pair is determined by itself rather than by specific query tokens, certain types of baselines should be included in the discussion, and, if possible, compared experimentally. InfiniPot [1] provides a training-free and query-agnostic method that aligns with chunked prefill KV cache eviction. Locret [2] also employs a training-based method to learn the importance score of each KV pair. These methods should be discussed and compared.

2. Although this paper focuses on evicting unimportant KV cache units during reasoning, its ability to process general long-context inputs should also be evaluated. Typical long-context benchmarks such as RULER [3] and LongBench [4] should be included. Eviction-based KV cache optimizations are not expected to perform well on difficult retrieval tasks in RULER (as shown in the Locret [2] paper), but this limitation should still be discussed in the paper to highlight potential weaknesses of the method.

3. The description of the training setup is unclear. The volume of training tokens, batch sizes, sequence lengths, and other hyperparameters should be included in the paper (these could be placed in the appendix). Note that another type of attention optimization: trainable sparse attention (e.g., NSA [5]), requires significantly more training data to achieve higher performance. Listing the training cost of TRIM-KV would help clarify the differences between these methods.

4. Is multi-turn conversation supported by TRIM-KV?

5. Writing should be improved. Appendix C is excessively long yet provides limited useful information. Figure 4 is also not providing much information (a smaller figure is preferred). The conclusion should be written in the authors’ own words rather than by including the model’s generated answers. The methods mentioned in Weakness 1 should also be incorporated into the related work section.

Overall, I appreciate this work and plan to increase my overall rating if the weaknesses and questions above are addressed.

---

[1] Infinipot: Infinite context processing on memory-constrained llms.

[2] Locret: Enhancing eviction in long-context llm inference with trained retaining heads.

[3] RULER: What's the Real Context Size of Your Long-Context Language Models?

[4] Longbench: A bilingual, multitask benchmark for long context understanding

[5] Native sparse attention: Hardware-aligned and natively trainable sparse attention

**Questions:**

See above.

---

> ### Author Response · Authors · 2025-11-20
> **Responses to Reviewer aGD2 [Part 1/2]**
>
> We thank the reviewer for taking the time to evaluate our paper and provide positive feedback. We address the reviewer’s remaining questions as follows.
>
> >  Q1. Comparison to InfiniPot and Locret.
>
> LocRet targets chunk prefilling for long-context inference, whereas our focus is long-horizon generation. Our training paradigm also differs: LocRet predicts attention logits per KV head, while we train retention gates end-to-end, allowing importance scores across heads to jointly optimize the eviction strategy. In inference, LocRet additionally relies on a hand-crafted sliding window (they call it a stabilizer) of size n_s = 3000 (about half the KV budget) for the latest chunk, and their results (Fig. 3) show that removing this heuristic substantially harms performance. In contrast, our method does not require any such heuristic; sliding-window–like behaviors emerge automatically from the learned policy, and some heads (e.g., layer 15, head 2 in Fig. 4) do not show any sliding windows at all. We are attempting to adapt their training strategy to the same setting as SeerAttn-R and our method for a fair comparison, but have been unsuccessful so far due to limited resources and training instability. We will update this thread if we can obtain reasonable performance for LocRet.
>
> Related to InfiniPot, we do agree that our paper will benefit from having one more query-agnostic baseline, so we will cover all three categories (learnable policy, attention-based policy, and query-agnostic policy). Unfortunately, we cannot find the public codebase for InfiniPot, and this work also focuses on chunked prefiling instead (although it can be adapted to reasoning, but its performance has not been verified). Therefore, we choose another baseline, KeyDiff, from NeuRIPS’25 as a representative for the query-agnostic baseline. The performance is worse than attention-based heuristics:
>
> | Method | 256  | 512  | 1024 | 2048 | 4096 | 32768 |
> |--------|------|------|------|------|------|-------|
> | FullKV | –    | –    | –    | –    | –    | 65.5  |
> | R-KV    | 0.2  | 1.9  | 18.5 | 42.7 | 63.6 | –     |
> | KeyDiff | 0.8  | 2.9  | 7.5  | 22.5 | 45.0 | –     |
> | TRIM-KV   | 37.5 | 61.5 | 70.0 | 72.3 | 74.0 | –     |
>
> R-KV is a more general method since it takes both diversity and attention score into consideration. We also want to emphasize that R-KV and KeyDiff are just accepted at NeuRips this year. and SeerAttn-R is a follow-up method of SeerAttn, which is also accepted at NeuRips this year. So we believe that baselines are solid, the most recent work.
>
> > Q2. Comparisons on Long-context tasks
>
> We chose LongMemEval for long-context evaluation because it more closely matches the natural use case of LLMs under very long but compressible contexts. As discussed in the SCBench paper, benchmarks like RULER and LongBench are designed for evaluating long-context performance under vanilla inference: they allow the model to see the full context and the query at compression time, and then repeatedly recompress the cache for different queries. This evaluation protocol does not reflect realistic usage of KV-cache compression methods, where the cache is typically compressed once and reused across queries. For this reason, we adopt the evaluation setting proposed by SCBench instead. The results with the KV budget 4096 are shown in the table below.
>
> | Method            | En.MultiChoice | Retr.KV | ICL.ManyShot | Math.Find | En.QA  | Code.RepoQA | En.Sum | Mix.Sum+NIAH | Retr.MultiHop |
> |-------------------|---------------:|--------:|-------------:|----------:|-------:|------------:|-------:|-------------:|--------------:|
> | Full KV           | 20.52          | 66.00   | 95.57        | 32.60     | 28.78  | 53.86       | 36.48  | 38.33        | 49.60         |
> | StreamingLLM | 5.68           | **2.20**| **100.00**   | 13.20     | 6.84   | 2.96        | 29.21  | 28.25        | 0.00          |
> | H2O          | 4.80           | 0.00    | **100.00**   | 8.00      | 3.70   | 0.46        | 8.97   | 6.51         | 0.27          |
> | SnapKV       | 10.04          | 0.00    | **100.00**   | **18.60** | 6.29  | 0.23        | 27.90  | 29.28        | 0.31          |
> | TRIM-KV      | **13.10**      | 0.00    | **100.00**   | 13.80     | **19.09** | **4.32** | **33.66** | **34.06** | **43.11**     |
>
> It can be seen that TrimKV can also be very effective for long-context compression scenarios for tasks that are compressible. All eviction methods fail to perform on difficult retrieval tasks, consistent with the observation from SCBench.

---

> ### Author Response · Authors · 2025-11-20
> **Responses to Reviewer aGD2 [Part 2/2]**
>
> > Q3. The description of the training setup.
>
> We describe our training setup in Appendix B.1, but agree it can be clarified further. Our settings largely follow SeerAttn-R to ensure a fair comparison, and we include these details for reproducibility.
>
> Regarding NSA, we are aware of this work, but it focuses on training LLMs from scratch with substantially modified attention dynamics, and its effectiveness at larger scales remains to be validated. In contrast, our method operates on existing LLMs and can turn any base model into a memory-bounded LLM without retraining from scratch, which we view as a key practical advantage.
>
> > Q4. Is multi-turn conversation supported by TRIM-KV?
>
> Yes. TRIM-KV is fully compatible with multi-turn conversation settings. In fact, our LongMemEval benchmark is designed to reflect this scenario. For each method, we enforce compression of the history from previous sessions before exposing the question in the next session. In other words, the compression method never sees future questions, so there is no information leakage. This setup closely reflects the real-world use case of KV compression methods in multi-turn conversations.
>
> > Q5. Writing should be improved. Appendix C and Figure 4 provide limited useful information. Related work section.
>
> Thanks for the suggestions. We added navigation in the caption of all figures.
>
> We would like to note that Figure 4 is discussed in Section 5.1, the emergent eviction heuristics paragraph.  Each subfigure is representative of one heuristic we found in our policy; some of them is surprising. For example, in layer 0, head 0, we can see that there are 5 co-existing sliding windows that emerged without any hard-coding.
>
> Appendix C is discussed in Section 5.1, interpretability paragraph. One of the most interesting observations from this is in Figure 22, where this head mostly keeps period tokens. This suggests that this head may implicitly perform gist tokens that summarize information from the previous sentence. This contrasts with recent trends that advocate for saving a chunk of tokens. Our results suggest that it can be more budget-efficient to save individual tokens because they already capture contextual information.
>
> We hope we have addressed all questions adequately and have revised the paper accordingly. Please let us know if any further clarifications would be helpful.
>
> Li, Yucheng, et al. "Scbench: A kv cache-centric analysis of long-context methods." arXiv preprint arXiv:2412.10319 (2024).

---

> ### Comment · Reviewer_aGD2 · 2025-11-20
>
> Thank you for your detailed and timely response.
>
> W1: I believe the discussion on the baselines is now sufficient.
>
> W2: The detailed scores provide strong evidence of TRIM-KV’s performance. The significant performance degradation observed with Retr.KV is very common for eviction-based methods, and it is good to see this limitation explicitly addressed in the discussion of the experimental results.
>
> W3: Thank you for providing additional details on training. I think the discussion on the difference between NSA and TRIM-KV should be included in the main paper.
>
> W4: Thank you for clarifying my question regarding multi-turn conversations.
>
> W5: The discussion on attention patterns is interesting. Thank you for including it in the appendix.
>
> Again, I appreciate your prompt and thorough response. One new issue: please do not alter the paper template. The header "Under review as a conference paper at ICLR 2026" should appear on every page. Once the discussion of W3 is added and the format is fixed, I will raise my overall rating.

---

> > ### Author Response · Authors · 2025-11-20
> > **Thank you!**
> >
> > We thank the reviewer for the prompt response and the continued constructive suggestions, which are crucial for improving our work.
> >
> > We have now added an explicit discussion of NSA and the Forgetting Transformer (FoX) in the related work section (lines 130–139), and thank you for pointing out the issue with the template.
> >
> > Overall, based on our preliminary results, we believe that, when properly trained on sufficiently general datasets, TRIM-KV could be a strong candidate to replace sink attention and sliding-window heuristics, which remain standard choices in current base models (e.g., sink attention in GPT-OSS and sliding windows in Qwen3).

---

> > > ### Comment · Reviewer_aGD2 · 2025-11-20
> > >
> > > Thanks for your update. I've increased my rating.

---

> ### Author Response · Authors · 2025-11-26
> **Added a comparison to LocRet**
>
> Dear Reviewer,
>
> We thank the reviewer very much for increasing the score. We would like to update on the comparison to LocRet, which we promised previously.
>
> Since we're still having a problem with achieving a reasonable performance for LocRet in our settings, which follow SeerAttention-R. To avoid reporting a result that could be an artifact from our implementation (LocRet also has many hyperparameters to tune), we decided to bring our method to chunked-prefill settings, which are targeted and verified by LocRet.
>
> To ensure a fair comparison, we use the exact settings (chunk size = 3072, budget = 6000) provided by LocRet and use their released checkpoint on Phi-3-mini-128K for evaluation. For TRIM-KV, we train our method on the LongAlpaca dataset, similar to LocRet, to make sure that our method does not benefit from the exposure to additional training datasets. The result on LongBench is reported in the following table,  together with the numbers from the LocRet paper (indicated by *):
>
> | Method | 2wikimqa | gov\_report | hotpotqa | lcc | multi\_news | mfieldqa | musique | narrativeqa | pssg\_count | pssg\_retrv | qasper | qmsum | repobench-p | samsum | triviaqa | Avg. $\Delta$ (%) |
> | :--- | :--- | :--- | :--- | :--- | :--- | :--- | :--- | :--- | :--- | :--- | :--- | :--- | :--- | :--- | :--- | :--- |
> | Full KV* | 33.37 | 33.35 | 43.06 | 51.86 | 26.57 | 49.82 | 19.82 | 18.21 | 2.97 | 93.50 | 41.07 | 19.51 | 58.02 | 23.15 | 86.38 | -- |
> | LocRet* | 35.93 | 33.46 | 48.70 | 52.61 | 26.41 | 52.77 | 25.12 | 24.56 | 3.00 | 83.00 | 40.17 | 23.35 | 57.16 | 26.37 | 82.39 | -- |
> | Full KV | *37.01* | *33.35* | *53.35* | **33.35** | 26.02 | **54.45** | *25.90* | **26.17** | **5.00** | **96.50** | **40.18** | **24.08** | 34.08 | **38.77** | **85.50** | 0.00 |
> | LocRet | **37.24** | 32.80 | 48.67 | 28.60 | **26.77** | *54.12* | **26.63** | 22.96 | *3.50* | 87.50 | 39.39 | 22.98 | *37.28* | *37.99* | 83.29 | -4.82 |
> | TRIM-KV | 36.65 | **33.37** | **54.78** | *33.08* | *26.02* | 53.25 | 25.39 | *25.00* | **5.00** | *94.50* | *40.17* | *23.59* | **37.46** | 36.82 | *83.38* | **-0.64** |
>
> Since the original LocRet's LongBench experiments have not been released, we use the default chat template of Phi-3-mini-128K for all runs. We observe discrepancies in the full-KV performance, which are likely attributable to differences in chat templates.
>
> Overall, TRIM-KV remains highly effective in the chunked-prefill setting. On average, TRIM-KV nearly matches full-KV performance, whereas LocRet incurs a 4.82% drop relative to full-KV cache inference. It is worth noting that LocRet relies on a heuristic stabilizer, which is basically a sliding window of size 2500, while our method requires no such heuristic. When we remove this stabilizer from LocRet, we observe a similar performance degradation to that reported in the original LocRet paper.
>
> We added a dedicated experiment section in Appendix B.3 for this chunked-prefill setting.
>
> It's worth noting that most existing KV cache eviction methods target one setting, long-generation, long-context understanding, or chunked prefilling, whereas TRIM-KV demonstrates advantages across all three. This highlights both the breadth and the extensiveness of our experimental evaluation.
>
> We hope that we have now cleared up all questions from the reviewer. Please let us know if there's anything that we can clarify.

---

> > ### Author Response · Authors · 2025-11-27
> > **Adding the experiment on LongBenchV2**
> >
> > Since LongBench is quite old and the context length of the evaluated samples is short, we add here the evaluation on LongBench-V2.
> >
> > | Method  | Acc. Short | Acc. Medium | Acc. Easy   | Acc. Hard   | Avg. Acc  | Δ (%)   |
> > |---------|-----------:|------------:|------------:|------------:|----------:|--------:|
> > | Full KV | *33.71*    | 18.60       | **34.44**   | *25.86*     | *28.79*   | *0.00*  |
> > | LocRet  | 32.02      | *19.78*     | *26.67*     | **28.74**   | 28.03     | -2.64   |
> > | TRIM-KV | **35.39**  | **20.93**   | **34.44**   | **28.74**   | **30.68** | **+6.56** |
> >
> > All settings are identical to those used in the previous LongBench experiment. Since the effective context length of Phi3-mini-128K is 128K tokens, we restrict evaluation to examples whose context length falls below this limit.
> >
> > Under this setup, TRIM-KV surpasses the Full KV baseline by 6.56% in average accuracy, highlighting the effectiveness of our method in long-context scenarios.

---

> > > ### Comment · Reviewer_aGD2 · 2025-11-27
> > >
> > > Thank you again for providing these extra results. These results fully show the contribution of TRIM-KV. I'm grateful for your time and effort during the response session.

---

### Official Review · Reviewer_dqKn · 2025-10-29

**Soundness:** 3
**Presentation:** 3
**Contribution:** 3
**Rating:** 8
**Confidence:** 4

**Summary:**

This paper proposes TRIM-KV, a learnable token retention mechanism for memory-bounded large language model (LLM) inference. Instead of relying on attention-based heuristics for KV-cache eviction, the authors introduce a lightweight retention gate that predicts each token’s intrinsic importance at creation time. The predicted retention score decays exponentially over time, inspired by Ebbinghaus’s forgetting curve, and governs which tokens are evicted when memory exceeds a fixed budget. The gates are trained via distillation from a frozen teacher LLM and a capacity regularization loss, ensuring negligible inference overhead. Experiments across multiple benchmarks — including GSM8K, MATH-500, AIME24, LongProc, and LongMemEval — demonstrate that TRIM-KV outperforms strong heuristic and learnable baselines under the same or even tighter memory budgets. Qualitative analyses further show that the learned retention scores align with intuitive heuristics such as sliding windows, sink tokens, and gist tokens, suggesting interpretability benefits.

**Strengths:**

1. Novel and well-motivated idea:
The paper identifies a fundamental limitation of attention-based eviction — that “recent attention ≠ importance” — and replaces it with a predictive, intrinsic importance estimation per token. This shift from reactive to proactive cache management is conceptually elegant and well-justified.
2. Brain-inspired design:
Modeling token importance decay via an exponential forgetting curve connects the method to cognitive science (Ebbinghaus), offering both theoretical grounding and interpretability.
3. Method simplicity and practicality:
TRIM-KV requires only a small MLP per layer/head, trained with frozen base weights. The added computation and memory are negligible, and inference remains fully compatible with Flash/FlexAttention kernels. The implementation is clear and deployable.
4. Comprehensive empirical validation:
Results are reported on five diverse benchmarks covering reasoning, long-context generation, and long-memory conversation. TRIM-KV consistently outperforms SeerAttn-R (learnable retrieval) and heuristic eviction methods (R-KV, SnapKV, H2O), sometimes even surpassing the full-cache baseline — a strong empirical claim rarely seen in this line of work.
5. Interpretability and analysis:
The authors visualize layer/head-wise retention scores, revealing emergent behaviors like sliding windows and sink tokens. This provides valuable insight into how LLMs allocate and forget contextual information.

**Weaknesses:**

Limited exploration of dynamic budgets:
The current method assumes a fixed memory budget M. It would be interesting to explore adaptive budgets (e.g., varying by layer, head, or task), especially since the authors mention this as future work.

**Questions:**

1. During inference, is the retention gate computed for every generated token, or cached and reused across time steps?
2. How sensitive is performance to the λ_cap hyperparameter and the initial bias b in the retention gate?

---

> ### Author Response · Authors · 2025-11-20
> **Responses to Reviewer dqKn**
>
> We thank the reviewer for taking the time to evaluate our paper and provide positive feedback. We address the reviewer’s remaining questions as follows.
>
> > Q1. Limited exploration of dynamic budgets
>
> In this paper, we focus on a fixed budget constraint for a fair comparison to existing baselines since they also use a fixed budget. Exploration using a dynamic budget for layers and heads is very interesting, but orthogonal to our work. We would expect that dynamic budget allocation using token retention would bring more gain compared to existing allocation methods that use attention heuristics since we focus on the intrinsic importance of tokens (query-anostic) while attention relies on query information, which makes it more difficult to compare across layers and heads.
>
> > Q2. During inference, is the retention gate computed for every generated token, or cached and reused across time steps?
>
> Token retention is only computed once for each token and cached alongside the key and value states. It incurs a minor increase in memory since each token retention is only a scalar value, which is only 1/d_kv of the key and value states (d_kv is the dimension of the key-value states). We discuss this in Appendix A1.
>
> > Q3. How sensitive is performance to the λ_cap hyperparameter and the initial bias b in the retention gate?
>
> Our training paradigm is robust to both hyperparameters. We fix lambda_cap = 1.0 and b=8 for all of our training and do not tune these values too much.
>
> The hinge-like loss is usually only active during the early training stage and will become zero when the capacity constraint is respected in the later training phase. So we keep lambda_cap = 1 for all of our training and do not tune this value.
>
> Related to the initial bias b, setting the initial bias too low makes the training difficult to converge since it deviates too much from the base model. Usually, we need a sufficiently high bias b to avoid too much forgetting during the early training stage. After crossing a threshold, the training is robust to this hyperparameter. We find that b = 8 is sufficient and fix it for all of our training, and higher values do not make much difference. We compared to b = 0 in Figure 9 in the appendix.
>
> We hope we have addressed all questions adequately and have revised the paper accordingly. Please let us know if any further clarifications would be helpful.

---

### Official Review · Reviewer_gKFH · 2025-11-01

**Soundness:** 2
**Presentation:** 3
**Contribution:** 3
**Rating:** 4
**Confidence:** 4

**Summary:**

The paper tackles KV cache limits for long context inference and proposes a learned decay gate to keep useful tokens.
The idea is easy to follow and fits well with current attention sparsity efforts.
In practice, it works reasonably well and the empirical bump over several benchmarks looks consistent.

However, I am not fully convinced the method is as conceptually “beyond attention” as the paper suggests. Even though the authors emphasize intrinsic token importance, the approach ultimately still plays through the attention path by reweighing the attention kernel.
It is not really removing attention as a relevance signal, but shifting when the weighting is introduced.
The learned score still modulates q·k, so the line between “intrinsic importance” and “attention-based importance” is blurrier than the paper presents.

On the training side, the method is positioned as a plug-in module that sits on top of frozen weights. If retention is so critical, it is natural to know why this is only applied for this scenario instead of making it part of the main training pipeline. Integrating the retention gates into pretraining or post training would likely change the model’s memory habits in a deeper way.

The model relies on RoPE or similar positional embedding, and the learned gate sits on top. The paper does not clarify whether this interacts with positional encodings. It might be fine in practice, but some comment or ablation on this point would strengthen the clarity of the claim that the model learns “intrinsic” retention.

Overall, the method seems useful and the results are good enough to merit attention.

**Strengths:**

1. Novel idea, fits cleanly into existing models, and works well without heavy changes.
2. Shows solid gains across tasks, occasionally even beating full-cache runs.
3. Lightweight enough to feel practical, not just academic.

**Weaknesses:**

Since it’s still a trained model, it raises the natural question of why this isn’t folded into normal model training, and the paper doesn’t address that integration path. It also doesn’t discuss how the learned decay interacts with positional encoding, which leaves some ambiguity around whether it’s learning true importance or just reinforcing a recency style bias.

**Questions:**

1. Since the method relies on training, why isn’t it integrated into the general training pipeline instead of being applied frozen models?
2. Does the interaction between the decay gate and existing positional encoding introduce redundancy or bias toward recency, and has this been tested or ablated?
3. If the model were trained from scratch with the retention mechanism active, would it converge to a different or stronger retention strategy than the post-hoc approach?
4. How does the system behave under much longer sequences or tool calling contexts?

---

> ### Author Response · Authors · 2025-11-20
> **Responses to Reviewer gKFH [Part 1/2]**
>
> We thank the reviewer for taking the time to evaluate our paper and especially for sharing your thoughts. We address the reviewer’s concerns point by point as follows.
>
> > Q1. Since it’s still a trained model, it raises the natural question of why this isn’t folded into normal model training instead of being applied to frozen models?
>
> We intentionally do not aim to replace standard attention in pretraining. Our problem setting is practical: given a strong pretrained LLM, how can we make its KV cache respect a strict memory budget by evicting unimportant tokens? This is already a real deployment concern, as evidenced by heuristic solutions such as sink attention (GPT-OSS) and sliding windows (e.g., Qwen3).
>
> Our contribution is a simple, learnable, and efficient plug-in that can be applied on top of frozen weights to turn an existing model into a memory-bounded one. This contrasts with approaches like Forgetting Transformer (Fox), which modify the attention mechanism itself and require heavy training from scratch; their effectiveness at large scale remains unclear, whereas our method can immediately leverage state-of-the-art pretrained LLMs, and our training with retention gates is very lightweight since it is no more than PEFT finetuning.
>
> We agree that integrating retention-gated attention into pretraining or post-training is an interesting direction that could shape the model’s `memory habits’ more deeply. For example, the model could better learn a way to coordinate the embeddings of the tokens in a more compressible way to facilitate the eviction. For example, the model might learn to coordinate token embeddings in a more compressible way to better support eviction, and gist tokens (observed in Figure 22 and discussed in Section 5.1.2, interpretability paragraph) might emerge more frequently. However, pursuing this direction would require substantial additional experimentation, architectural design, and computational resources, which we consider beyond the scope of this work.
>
> Overall, we believe the potential to replace the standard attention should be considered as our strength instead of a weakness, but we agree that a discussion of future work would significantly benefit our paper, and we included a discussion in the future work section.
>
> > Q2.1 I am not fully convinced the method is as conceptually “beyond attention” as the paper suggests… the line between “intrinsic importance” and “attention-based importance” is blur.
>
> Our claim is not that we abandon attention, but that we separate two distinct notions of importance: short-term, query-dependent attention vs long-term, query-agnostic retention for eviction.
>
> Standard attention scores are query-dependent and recomputed at every step. They answer “how much should token i matter for this current query?” and thus capture short-term, myopic utility for predicting the next token.
>
> KV eviction, however, is a long-horizon decision: once a token is evicted, it can never affect any future step. A good eviction policy should therefore depend on a token’s intrinsic long-term utility, how useful it is expected to be over the rest of the sequence, and how long it has already stayed in the cache, not just on the current query.
>
> Our retention gate is designed exactly for this role. The decision variable $\alpha_{ti}$ (equation 1, Section 3.2) is introduced to the standard attention to determine whether token i is still present in the cache: it starts at 1 and abruptly drops to 0 as the token is evicted. But this is binary and cannot be learned. So, token retention is a smooth approximation of this behavior to make the whole process learnable. We added a dedicated paragraph to the paper to make this clear.
>
> > Q2.2. Why do we need to plug in to modulate the attention score? “The learned score still modulates q·k, so the line between “intrinsic importance” and “attention-based importance” is blurrier than the paper presents.
>
> Token retention will be useless if it does not correlate with the actual attention contribution of a token in the future steps. The purpose of retention-gated attention is to create a *proxy* mechanism that approximates the standard attention running with eviction during inference. So that we can optimize this proxy mechanism to mimic the behavior of the full attention and respect the memory constraint.
>
> After training finished, we do not need retention-gated attention anymore, and during inference, we use the standard attention running with eviction decision guided by the retention scores.
>
> We would like to emphasize again here that we do not aim use use retention-gated attention to replace the standard attention. Instead, we create a learnable proxy mechanism to approximate the standard attention with eviction. There is a potential to replace the standard attention, but it will need to be investigated thoroughly before making any claim, since such a claim has a consequential impact, as most of the SOTA architecture relies on the attention.

---

> ### Author Response · Authors · 2025-11-20
> **Responses to Reviewer gKFH [Part 2/2]**
>
> > Q3. Does the interaction between the decay gate and existing positional encoding introduce redundancy or bias toward recency, and has this been tested or ablated?
>
> Our retention mechanism is designed to be positional-encoding agnostic and does not add any extra recency bias beyond what is already present in the base model.
>
> The exponential decay in the retention-gated attention is a smooth approximation of the decay process from 1 to 0 of $\alpha$ in the standard attention with eviction, not to encode the positional information. Regardless of whether the model uses absolute positions, RoPE, or no positional encoding, that information is already folded into x, q, and k in the standard attention (Eq. (1) in our paper). At inference time, we still use standard attention; the learned decay only models how a token’s retention decreases over time for eviction.
>
> Implementation-wise, when using RoPE, we follow prior work (R-KV, SnapKV) and cache post-rotated keys, so the eviction is orthogonal to the positional embeddings used. We included this discussion in the revision for clarity.
>
> > Q4. If the model were trained from scratch with the retention mechanism active, would it converge to a different or stronger retention strategy than the post-hoc approach?
>
> We expect that joint training from scratch could lead to a different or even stronger retention strategy, but we would refrain from making any claim here.
>
> Our current setup freezes the base model and only trains the retention gate, which makes optimization stable and keeps the method practical for existing large LLMs. If we also updated the base model under a strict memory budget, capacity loss and training stabilities would need to be very carefully managed, and demonstrating good scaling behavior at that regime would require substantial additional computation and engineering. Further, when using retention-gated attention during inference, the catastrophic forgetting also needs to be evaluated since it changes the attention dynamic a lot.
>
> > Q5. How does the system behave under much longer sequences or tool calling contexts?
>
> We evaluate TrimKV on very long contexts in both LongMemEval (up to 123k tokens) and SCBench (128k tokens), as reported in Section 5.2. The results for SCBench with a budget of 4096 are shown in the table below.
>
> | Method            | En.MultiChoice | Retr.KV | ICL.ManyShot | Math.Find | En.QA  | Code.RepoQA | En.Sum | Mix.Sum+NIAH | Retr.MultiHop |
> |-------------------|---------------:|--------:|-------------:|----------:|-------:|------------:|-------:|-------------:|--------------:|
> | Full KV           | 20.52          | 66.00   | 95.57        | 32.60     | 28.78  | 53.86       | 36.48  | 38.33        | 49.60         |
> | StreamingLLM | 5.68           | **2.20**| **100.00**   | 13.20     | 6.84   | 2.96        | 29.21  | 28.25        | 0.00          |
> | H2O          | 4.80           | 0.00    | **100.00**   | 8.00      | 3.70   | 0.46        | 8.97   | 6.51         | 0.27          |
> | SnapKV       | 10.04          | 0.00    | **100.00**   | **18.60** | 6.29  | 0.23        | 27.90  | 29.28        | 0.31          |
> | TRIM-KV      | **13.10**      | 0.00    | **100.00**   | 13.80     | **19.09** | **4.32** | **33.66** | **34.06** | **43.11**     |
>
> As shown in the table, TrimKV remains highly effective in compressible long-context settings. All eviction methods, including ours, struggle with the most difficult retrieval tasks where context is incompressible. This is consistent with the observation from SCBench.
>
> Tool-calling is a natural application of our method, and we expect TrimKV to be advantageous there. However, we were not able to run dedicated experiments during the rebuttal period due to limited compute, the need for large models (e.g., on ComplexFuncBench, only 72B+ or closed-source models reach a reasonable performance), and suitable tool-use training data. While Figure 7 shows that retention gates trained on general data can transfer to math tasks, we still expect task-relevant data to be beneficial. We now discuss tool-calling and this extension more explicitly in the future work section.
>
> We hope we have addressed all questions adequately and have revised the paper accordingly. Please let us know if any further clarifications would be helpful.
>
> Li, Yucheng, et al. "Scbench: A kv cache-centric analysis of long-context methods." arXiv preprint arXiv:2412.10319 (2024).

---

### Official Review · Reviewer_dB8X · 2025-11-01

**Soundness:** 2
**Presentation:** 2
**Contribution:** 2
**Rating:** 4
**Confidence:** 4

**Summary:**

This paper proposes TRIM-KV, a trainable approach for KV cache eviction that assigns intrinsic importance scores to tokens at creation time through lightweight retention gates. The method trains these gates using distillation and capacity losses to enable memory-bounded inference while preserving model quality.

**Strengths:**

1. The paper introduces a trainable approach to KV cache eviction that learns token-level importance scores through retention gates. This design allows the model to identify and retain intrinsically important tokens
2. The authors conduct experiments across multiple mathematical reasoning tasks and show competitive results.

**Weaknesses:**

1. The paper lacks comparison with LocRet, a highly relevant recent work that also computes importance scores for each token and discards low-importance tokens to reduce memory overhead.

Locret: Enhancing eviction in long-context LLM inference with trained retaining heads

2. The paper would benefit from comprehensive evaluation on established long-context understanding benchmarks such as RULER and LongBench-V2 to validate the method's effectiveness.

3. The use of exponential decay for computing token importance scores appears problematic. Even with an initial retention score of 0.999, the importance decays to less than 5% after approximately 3000 tokens. This aggressive decay suggests the model may essentially degenerate into a sliding window approach for longer contexts.

**Questions:**

Please refer to Weaknesses.

---

> ### Author Response · Authors · 2025-11-20
> **Responses to Reviewer dB8X [Part 1/2]**
>
> We thank the reviewer for taking the time to evaluate our paper. We address the reviewer’s concerns by points as follows.
>
> > Q1. Comparison to LocRet.
>
> We thank the reviewer for the suggestion. LocRet targets chunk prefilling for long-context inference, whereas our focus is long-horizon generation. Our training paradigm also differs: LocRet predicts attention logits per KV head, while we train retention gates end-to-end, allowing importance scores across heads to jointly optimize the eviction strategy. In inference, LocRet additionally relies on a hand-crafted sliding window (they call it a stabilizer) of size n_s = 3000 (about half the KV budget) for the latest chunk, and their results (Fig. 3) show that removing this heuristic substantially harms performance. In contrast, our method does not require any such heuristic; sliding-window–like behaviors emerge automatically from the learned policy, and some heads (e.g., layer 15, head 2 in Fig. 4) do not show any sliding windows at all. We are attempting to adapt their training strategy to the same setting as SeerAttn-R and our method for a fair comparison, but have been unsuccessful so far due to limited resources and training instability. We included this discussion in our revision in Appendix A.3. We will update this thread if we can obtain reasonable performance for LocRet.
>
> Regarding baselines, we believe we cover the most important and recent methods. We added KeyDiff, a query-agnostic baseline, as suggested by Reviewer aGD216. R-KV and KeyDiff were both accepted to NeurIPS25, and SeerAttn-R is a follow-up to the NeurIPS25-accepted SeerAttn paper. SeerAttn-R is a strong learnable KV-retrieval baseline that avoids eviction at the cost of increased I/O overhead; as noted by SCBench, KV-retrieval methods typically enjoy performance advantages over eviction-based methods because they do not discard tokens. This makes our improvements over SeerAttn-R particularly meaningful. Overall, we believe our baselines are state-of-the-art.
>
> > Q2. Evaluation on long-context understanding benchmarks
>
> We chose LongMemEval for long-context evaluation because it more closely matches the natural use case of LLMs under very long but compressible contexts. As discussed in the SCBench paper, benchmarks like RULER and LongBench are designed for evaluating long-context performance under vanilla inference: they allow the model to see the full context and the query at compression time, and then repeatedly recompress the cache for different queries. This evaluation protocol does not reflect realistic usage of KV-cache compression methods, where the cache is typically compressed once and reused across queries. For this reason, we adopt the evaluation setting proposed by SCBench instead. The results with budget 4096 are shown in the table below.
>
> | Method            | En.MultiChoice | Retr.KV | ICL.ManyShot | Math.Find | En.QA  | Code.RepoQA | En.Sum | Mix.Sum+NIAH | Retr.MultiHop |
> |-------------------|---------------:|--------:|-------------:|----------:|-------:|------------:|-------:|-------------:|--------------:|
> | Full KV           | 20.52          | 66.00   | 95.57        | 32.60     | 28.78  | 53.86       | 36.48  | 38.33        | 49.60         |
> | StreamingLLM | 5.68           | **2.20**| **100.00**   | 13.20     | 6.84   | 2.96        | 29.21  | 28.25        | 0.00          |
> | H2O          | 4.80           | 0.00    | **100.00**   | 8.00      | 3.70   | 0.46        | 8.97   | 6.51         | 0.27          |
> | SnapKV       | 10.04          | 0.00    | **100.00**   | **18.60** | 6.29  | 0.23        | 27.90  | 29.28        | 0.31          |
> | TRIM-KV      | **13.10**      | 0.00    | **100.00**   | 13.80     | **19.09** | **4.32** | **33.66** | **34.06** | **43.11**     |
>
> As shown in the table, TrimKV remains highly effective in compressible long-context settings. All eviction methods, including ours, struggle with the most difficult retrieval tasks where context is incompressible. This is consistent with the observation from SCBench.

---

> > ### Author Response · Authors · 2025-11-20
> > **Responses to Reviewer dB8X [Part 2/2]**
> >
> > > The use of exponential decay. Even with an initial retention score of 0.999, the importance decays to less than 5% after approximately 3000 tokens. This aggressive decay suggests the model may essentially degenerate into a sliding window approach for longer contexts.
> >
> > First, 0.999 is actually a low retention score in our setting: it is just the initial value induced by the bias (b=8) (i.e., ($\sigma(8)\approx 0.9996$) before learning. Even at this level, ($0.9996^{3000} \approx 30\\%$), so small changes in the learned retention score have a large effect over long horizons. For genuinely important tokens, the network can assign much larger sigmoid logits. For example, with (logit=20), the per-step retention ($\sigma(20)$) is so close to 1 that after 128k steps the cumulative retention is still about 99.7%. Thus, whether a token "slides out" or persists is governed by its learned importance, not by an inherently aggressive decay.
> >
> > Numerically, this is stable because we operate entirely in log space: we track $(t-i)\log \beta_i$ rather than explicitly materializing very high-precision retention values.
> >
> > Empirically, our LongMemEval experiments use contexts of up to 123k tokens, where our method significantly outperforms simple sink attention and sliding-window baselines (StreamingLLM), confirming robustness at long context lengths.
> >
> > Finally, the effective “aggressiveness” of the sigmoid logits is learned from data and depends on the value budget M used during training. For example, the average log retention of a model trained on math-220k (1k–16k tokens) is −0.21 (\~81% retention per step), whereas for a model trained on synthlong (>32k tokens) it is −0.0057 (\~99.4% per step). Note that for eviction, what matters most is the relative importance across tokens, rather than the absolute retention values.
> >
> > We hope that we have addressed all questions adequately. We revised the paper accordingly. Please let us know if there are any other clarifications we could make.
> >
> > Li, Yucheng, et al. "Scbench: A kv cache-centric analysis of long-context methods." arXiv preprint arXiv:2412.10319 (2024).

---

> ### Author Response · Authors · 2025-11-26
> **Adding a comparison to LocRet**
>
> Dear Reviewer,
>
> Since we're still having a problem with achieving a reasonable performance for LocRet in our settings, which follow SeerAttention-R. To avoid reporting a result that could be an artifact from our implementation (LocRet also has many hyperparameters to tune), we decided to bring our method to chunked-prefill settings, which are targeted and verified by LocRet.
>
> To ensure a fair comparison, we use the exact settings (chunk size = 3072, budget = 6000) provided by LocRet and use their released checkpoint on Phi-3-mini-128K for evaluation. For TRIM-KV, we train our method on the LongAlpaca dataset, similar to LocRet, to make sure that our method does not benefit from the exposure to additional training datasets. The result on LongBench is reported in the following table,  together with the numbers from the LocRet paper (indicated by *, bold indicates the best method, and italic indicates the second-best method):
>
> | Method | 2wikimqa | gov\_report | hotpotqa | lcc | multi\_news | mfieldqa | musique | narrativeqa | pssg\_count | pssg\_retrv | qasper | qmsum | repobench-p | samsum | triviaqa | Avg. $\Delta$ (%) |
> | :--- | :--- | :--- | :--- | :--- | :--- | :--- | :--- | :--- | :--- | :--- | :--- | :--- | :--- | :--- | :--- | :--- |
> | Full KV* | 33.37 | 33.35 | 43.06 | 51.86 | 26.57 | 49.82 | 19.82 | 18.21 | 2.97 | 93.50 | 41.07 | 19.51 | 58.02 | 23.15 | 86.38 | -- |
> | LocRet* | 35.93 | 33.46 | 48.70 | 52.61 | 26.41 | 52.77 | 25.12 | 24.56 | 3.00 | 83.00 | 40.17 | 23.35 | 57.16 | 26.37 | 82.39 | -- |
> | Full KV | *37.01* | *33.35* | *53.35* | **33.35** | 26.02 | **54.45** | *25.90* | **26.17** | **5.00** | **96.50** | **40.18** | **24.08** | 34.08 | **38.77** | **85.50** | 0.00 |
> | LocRet | **37.24** | 32.80 | 48.67 | 28.60 | **26.77** | *54.12* | **26.63** | 22.96 | *3.50* | 87.50 | 39.39 | 22.98 | *37.28* | *37.99* | 83.29 | -4.82 |
> | TRIM-KV | 36.65 | **33.37** | **54.78** | *33.08* | *26.02* | 53.25 | 25.39 | *25.00* | **5.00** | *94.50* | *40.17* | *23.59* | **37.46** | 36.82 | *83.38* | **-0.64** |
>
> Since the original LocRet's LongBench experiments have not been released, we use the default chat template of Phi-3-mini-128K for all runs. We observe discrepancies in the full-KV performance, which are likely attributable to differences in chat templates.
>
> Overall, TRIM-KV remains highly effective in the chunked-prefill setting. On average, TRIM-KV nearly matches full-KV performance, whereas LocRet incurs a 4.82% drop relative to full-KV cache inference. It is worth noting that LocRet relies on a heuristic stabilizer, which is basically a sliding window of size 2500, while our method requires no such heuristic. When we remove this stabilizer from LocRet, we observe a similar performance degradation to that reported in the original LocRet paper.
>
> We added a dedicated experiment section in Appendix B.3 for this chunked-prefill setting.
>
> It's worth noting that most existing KV cache eviction methods target one setting, long-generation, long-context understanding, or chunked prefilling, whereas TRIM-KV demonstrates advantages across all three. This highlights both the breadth and the extensiveness of our experimental evaluation.
>
> We hope that we have now addressed all questions raised by the reviewer. Please let us know if there's anything that we can clarify.

---

> > ### Author Response · Authors · 2025-11-27
> > **Adding the comparison on LongBenchV2**
> >
> > Since LongBench is quite old and the context length of the evaluated samples is short, we add here the evaluation on LongBench-V2 as suggested by the reviewer.
> >
> > | Method  | Acc. Short | Acc. Medium | Acc. Easy   | Acc. Hard   | Avg. Acc  | Δ (%)   |
> > |---------|-----------:|------------:|------------:|------------:|----------:|--------:|
> > | Full KV | *33.71*    | 18.60       | **34.44**   | *25.86*     | *28.79*   | *0.00*  |
> > | LocRet  | 32.02      | *19.78*     | *26.67*     | **28.74**   | 28.03     | -2.64   |
> > | TRIM-KV | **35.39**  | **20.93**   | **34.44**   | **28.74**   | **30.68** | **+6.56** |
> >
> > All settings are identical to those used in the previous LongBench experiment. Since the effective context length of Phi3-mini-128K is 128K tokens, we restrict evaluation to examples whose context length falls below this limit.
> >
> > Under this setup, TRIM-KV surpasses the Full KV baseline by 6.56% in average accuracy, highlighting the effectiveness of our method in long-context scenarios.

---

### Author Response · Authors · 2025-11-20
**General Response**

We thank all the reviewers for spending the time to evaluate our paper and provide valuable feedback to improve our paper. In the following, we summarize the changes we have made during the rebuttal:

- We included an experiment on SCBench, a recent benchmark to evaluate the long-context performance of KV cache compression methods under realistic settings, as suggested by Reviewer dB8X, gKFH, and aGD2.
- We added discussions to LocRet and InfiniPot as suggested by Reviewer dB8X and aGD2.
- We add a comparison to KeyDiff, a query-agnostic baseline as suggested by Reviewer aGD2.
- We improved the future work discussion on the potential use of retention-gated attention to replace the standard attention and train the transformer with retention-gated attention from scratch or post-training, as suggested by Reviewer gKFH.
- We improved the writing to clarify some points made by the Reviewer gKFH and aGD2.

Overall, we believe our work is in a strong position relative to contemporaneous literature:

- **Novelty-wise**: The central contribution of this work is a shift in perspective from attention-based importance (short-term, query-dependent) to token-retention importance (long-term, query-agnostic). While query-agnostic importance itself is not new, our formulation revisits attention with eviction by introducing the eviction decision variable $\alpha_{ti}$ and its smoothed exponential decay $\beta_i^{t-i}$, yielding an effective training strategy that is no more than parameter-efficient finetuning yet works well for long-horizon generation. We also view our findings on emergent eviction policies and interpretability as noteworthy: in contrast to recent trends that favor chunk-wise caching, our results suggest it may be more budget-effective to retain individual tokens, which already encode rich contextual information.
- **Experimentation-wise**: we believe our experimental setup is solid, as our baselines consist largely of recent NeurIPS-accepted methods. The gains over SeerAttn-R, a learnable KV-retrieval approach, are particularly significant, since KV-retrieval typically enjoys a performance advantage at the cost of I/O overhead by avoiding eviction permanently. Moreover, most prior work focuses on either the prefilling phase or decoding phase, but rarely both; our experiments demonstrate benefits in both regimes. The following are baselines and benchmarks that are evaluated:
    - Baselines: `R-KV (NIPS25)`, `SeerAttn-R (Arxiv-June'25)`, `SnapKV(NIPS'24)`, `StreamingLLM (ICLR'24)`, `H2O (NIPS'23)`, `KeyDiff (NIPS25)`, `LocRet (Arxiv-Oct'24)`
    - Benchmarks: `Math (GSM8K, MATH-500, AIME24)`, `LongProc`, `LongMemEval`, `SCBench`, `LongBench`, `LongBenchV2`.

     Compared to baselines and benchmarks used in the literature of this line of work, we believe this is extensive.
- **Application-wise**: Our method is general, simple, and efficient, and can be plugged into existing pretrained LLMs without architectural changes or base parameter updates. As such, we believe that, when trained on sufficiently broad data, TRIM-KV is a strong candidate to replace heuristic mechanisms such as sink attention and sliding windows, which remain standard in current base models (e.g., sink attention in GPT-OSS and sliding windows in Qwen3).

Best,
Authors

---

> ### Author Response · Authors · 2025-11-26
> **An additional update**
>
> We would like to update that we added an experiment to compare with LocRet in the chunked-prefilling setting, which is targeted by LocRet. This result demonstrates that TRIM-KV provides advantages in all three settings: long-generation, long-context understanding, or chunked prefilling, highlighting both the breadth and the extensiveness of our experimental evaluation.
>
> Edit: We have now added evaluation on LongBenchV2. The result of TRIM-KV surpasses the full KV cache inference in this benchmark.

---

### Author Response · Authors · 2025-12-03
**Summary**

Dear ACs, SACs, and reviewers,

Thank you for the time and effort you have devoted to evaluating our submission and participating in the discussion. Due to an unfortunate incident this year, the discussion phase was terminated earlier than expected, and reviewers are unable to engage in further discussions. For conciseness, we summarize below the main points that appear to be shared across the reviews received so far.

From initial reviews:
- **Novelty**: Our method was described as “elegant, novel, and well-justified” by Reviewer dqKn, “reasonable, effective, and novel” by Reviewer aGD2, and a “novel idea [that] fits cleanly to existing model[s]” by Reviewer gKFH.
- **Practicality**: The approach was characterized as “simple, clear, and deployable” by Reviewer dqKn, offering “significant speedups” according to Reviewer aGD2, and “lightweight enough to feel practical, not just academic” by Reviewer gKFH.
- **Performance gains**: Reviewers highlighted “a strong empirical claim rarely seen in this line of work” (Reviewer dqKn), “solid gains across tasks” (Reviewer gKFH), “strong evidence of TRIM-KV’s performance” (Reviewer aGD2), and “competitive results” (Reviewer dB8X).
- **Extensiveness of the evaluation**: The experiments were described as comprehensive by Reviewers dqKn and aGD2, and as demonstrating “gains across multiple tasks” by Reviewers gKFH and dB8X.

Several concerns and questions were addressed during the rebuttal period:
- **Reviewer dB8X**: The reviewer requested comparisons with LocRet and additional experiments on long-context understanding tasks, which we have included. They also raised a concern about the exponential decay form of the retention scores; we clarified that this choice is reasonable and well-motivated both empirically and numerically.
- **Reviewer gKFH**: The reviewer’s main questions focused on the post-hoc application of our method to frozen LLMs and the possibility of fully replacing standard attention with retention-gated attention. We clarified the position of our work and the practical advantages of our setting, which allows our method to be directly applied on top of existing LLMs. This is a key benefit over other sparse attention models, such as FoX and NSA, which require substantial training from scratch and whose effectiveness at large scale therefore remains unverified (as also noted by Reviewer aGD2). We view the potential to replace standard attention as promising future work (one that would demand significant design effort and computational resources) and believe this potential should be regarded as an additional strength of our proposal.
- **Reviewer dqKn**: The reviewer had questions regarding the dynamic budget mechanism and hyperparameter tuning, which we clarified in our rebuttal.
- **Reviewer aGD2**: The reviewer requested comparisons with LocRet and an additional query-agnostic baseline, as well as an evaluation on long-context understanding tasks. We have incorporated all of these requested experiments. After the rebuttal, the reviewer increased the score to 8.

We hope this summary is helpful.

Best,
Authors

---

### Meta-Review · Area_Chair_7hqh · 2026-01-07

**Summary:**

The paper proposes TRIM-KV, a trainable retention-gate for KV-cache eviction that estimates intrinsic token importance at creation time and decays it over time. The rebuttal clarifies scope (eviction vs retrieval/selection), adds recent baselines (KeyDiff, SeerAttn-R), and supplies results on SCBench, LongMemEval, LongBench-V2, and chunked-prefill settings matching LocRet. It argues that exponential decay is learned and not inherently aggressive, and that retention is computed once per token with negligible overhead. Empirically, TRIM-KV is competitive or superior across compressible long-context tasks and long CoT, sometimes approaching or surpassing full-KV; it struggles on hard retrieval like other eviction methods. Conceptual novelty is moderate but practical impact is clear. Presentation improved; training details and formatting were addressed.

**Reviewer Concerns:**

Addressed: lack of LocRet and query-agnostic baselines; added KeyDiff and a careful LocRet comparison in its native chunked-prefill regime using their checkpoint and settings. Added long-context evaluations beyond reasoning (SCBench, LongMemEval, LongBench-V2), multi-turn support, chunked-prefill results, and sensitivity/implementation details (λ_cap, bias b, log-space stability). Clarified the role of retention-gated attention as a proxy during training only, with standard attention at inference plus eviction. Clarified positional encoding interactions and that retention is query-agnostic and cached per token.

Outstanding: Conceptual positioning remains somewhat incremental relative to prior output/retention-aware approaches; the distinction from attention-based importance is argued but still intertwined via attention modulation during training. A unified, centralized efficiency analysis (token-level per-step overhead, end-to-end throughput across context lengths and serving regimes) could be more concise in the main text. Breadth on ultra-long, hard retrieval remains limited (consistent with the method class), and integration into pretraining is left as future work.

**Reviewer Scores:**

R1 (original 4): Likely to 5 after added baselines, SCBench, chunked-prefill, and clarifications, though novelty concerns persist.

R2 (original 4): Likely to 5 given clearer scope, proxy-vs-inference story, and positional encoding discussion; still cautious on conceptual separation from attention.

R3 (original 6): Likely to 6–7; appreciates completeness and practicality; minor clarity issues remain.

R4 (original 8): Stays at 8; requested items (α/decay sensitivity, added analysis) addressed; positive on empirical breadth.

---

### Decision · Program_Chairs · 2026-01-26

Accept (Poster)